# A transcriptomic atlas of *Aedes aegypti* reveals detailed functional organization of major body parts and gut regional specializations in sugar-fed and blood-fed adult females

**Bretta Hixson, Xiao-Li Bing[†], Xiaowei Yang[‡], Alessandro Bonfini, Peter Nagy, Nicolas Buchon***

Cornell Institute of Host-Microbe Interactions and Disease, Department of Entomology, Cornell University, Ithaca, United States

**\*For correspondence:**
nicolas.buchon@cornell.edu

**Present address:** [†]Department of Entomology, Nanjing Agricultural University, Jiangsu, China; [‡]State Key Laboratory for Biology of Plant Diseases and Insect Pests, Institute of Plant Protection, Chinese Academy of Agricultural Sciences, Beijing, China

**Competing interest:** The authors declare that no competing interests exist.

**Abstract** Mosquitoes transmit numerous pathogens, but large gaps remain in our understanding of their physiology. To facilitate explorations of mosquito biology, we have created Aegypti-Atlas (http://aegyptiatlas.buchonlab.com/), an online resource hosting RNAseq profiles of Ae. aegypti body parts (head, thorax, abdomen, gut, Malpighian tubules, ovaries), gut regions (crop, proventriculus, anterior and posterior midgut, hindgut), and a gut time course of blood meal digestion. Using Aegypti-Atlas, we provide insights into regionalization of gut function, blood feeding response, and immune defenses. We find that the anterior and posterior midgut possess digestive specializations which are preserved in the blood-fed state. Blood feeding initiates the sequential induction and repression/depletion of multiple cohorts of peptidases. With respect to defense, immune signaling components, but not recognition or effector molecules, show enrichment in ovaries. Basal expression of antimicrobial peptides is dominated by holotricin and gambicin, which are expressed in carcass and digestive tissues, respectively, in a mutually exclusive manner. In the midgut, gambicin and other effectors are almost exclusively expressed in the anterior regions, while the posterior midgut exhibits hallmarks of immune tolerance. Finally, in a cross-species comparison between Ae. aegypti and Anopheles gambiae midguts, we observe that regional digestive and immune specializations are conserved, indicating that our dataset may be broadly relevant to multiple mosquito species. We demonstrate that the expression of orthologous genes is highly correlated, with the exception of a 'species signature' comprising a few highly/disparately expressed genes. With this work, we show the potential of Aegypti-Atlas to unlock a more complete understanding of mosquito biology.

## Editor's evaluation

Hixson et al. provide a large overview of gene expression level of the mosquito Aedes aegypti through the use of RNA-seq. They analyse gene expression changes in the digestive tract, as well as the 3 body regions and the ovaries in various conditions. These organ-specific transcriptomes fill a hole in our understanding of mosquito vector biology and will be an excellent starting point for many researchers to produce new projects.

## Introduction

Hematophagous mosquito vectors contribute substantially to the world's disease burden through the transmission of *Plasmodium* parasites and arboviruses ("*World Health Organisation, 2020a* | Vector-borne diseases,"). With the rise of insecticide resistance (*Moyes et al., 2017*; *Ranson et al., 2011*; "WHO | Insecticide resistance," *World Health Organisation, 2020b*), and the expansion of mosquito species' ranges potentiated by climate change (*Liu et al., 2020*), new and more efficient means of vector control are needed, grounded in a solid understanding of mosquito physiology. The FlyAtlas project (*Chintapalli et al., 2007*; *Leader et al., 2018*) contributed to the field of insect physiology, not only by characterizing the functions associated with the major body parts and organs of *Drosophila melanogaster*, but by making it possible to quickly ascertain where in the body, and in what quantity, any gene of interest is transcribed. FlyAtlas also showed how the various contributions of different organs added up to constitute the whole fly transcriptome, providing an estimate of the relative transcriptional yield of the parts of the *Drosophila melanogaster* body. Similar anatomical datasets have been created for several species of *Anopheles* (*Baker et al., 2011*; *Martínez-Barnetche et al., 2012*; *Sreenivasamurthy et al., 2017*), but not for *Aedes* mosquitoes. While transcriptional profiles have been created for various *Ae. aegypti* body parts in diverse studies (*e.g* head (*Ptitsyn et al., 2011*), Malpighian tubules (*Li et al., 2017*), midgut (*Angleró-Rodríguez et al., 2017*; *Cui and Franz, 2020*; *Dong et al., 2017*; *Hyde et al., 2020*; *Liu et al., 2016*; *Raquin et al., 2017*; *Xiao et al., 2017*), ovaries (*Akbari et al., 2013*; *Nag et al., 2021*), carcass (*Akbari et al., 2013*; *Choi et al., 2012*), the lack of uniformity of strain and methodology between these studies makes direct comparison difficult.

The mosquito gut is of special interest to vector control, as it is central to the hematophagous lifestyle and serves as the first interface between a mosquito and the pathogens it transmits. Insect guts are highly regionalized organs, with specialized functions differentially distributed along their length (*Benguettat et al., 2018*; *Buchon and Osman, 2015*; *Buchon et al., 2013*; *Dutta et al., 2015*; *Lemaitre and Miguel-Aliaga, 2013*). The mosquito gut is divided into five anatomically distinct regions: foregut (comprising pharynx, dorsal diverticula, and crop), proventriculus, anterior midgut, posterior midgut, and hindgut. The crop and dorsal diverticula are muscular cuticle-lined sacs which store imbibed sugar for gradual release into the midgut (*Clements, 1992*). In *Anopheles gambiae*, the proventriculus and anterior midgut were found to exhibit an enrichment of transcripts encoding antimicrobial peptides (AMPs) and anti-*Plasmodium* factors (*Warr et al., 2007*) suggesting that these regions serve a special defensive function against orally acquired pathogens. Within the midgut, digestive functions are believed to be divided between the anterior and posterior regions, with the former specializing in the digestion of nectar but playing no direct role in the digestion of the blood meal, which is processed in the posterior midgut (*Hecker, 1977*). The hindgut receives and eliminates excreta from the midgut and urine from the Malpighian tubules (*Clements, 1992*). Genome-wide transcriptomic profiles have the potential to reveal much more detailed information about the regionalization of function in the mosquito gut. In recent years, many groups have used microarrays and RNAseq to create transcriptomic profiles of mosquito midguts, mostly in the context of infection (*Baker et al., 2011*; *Cui and Franz, 2020*; *Dong et al., 2017*; *Hyde et al., 2020*; *Raquin et al., 2017*; *Vedururu et al., 2019*; *Vlachou et al., 2005*; *Warr et al., 2007*). However, to our knowledge, only a single microarray-based study has ever characterized the transcriptomes of individual mosquito midgut regions (*Warr et al., 2007*) and no transcriptomic profiles have been generated for either foregut or hindgut tissues in any mosquito species.

The *Ae. aegypti* gut's response to the ingestion of a blood meal has been characterized as a biphasic process. In the first phase, mRNAs encoding a cohort of 'early' serine endopeptidases, transcribed in response to juvenile hormone (JH) secretion during previtellogenic maturation, are translated (*Bian et al., 2008*; *Felix et al., 1991*; *Jiang et al., 1997*; *Noriega et al., 1996*; *Noriega et al., 1997*; *Noriega et al., 2001*). In the second, ecdysone signaling mediates the transcription of a 'late' cohort of endo and exopeptidases, with the transcripts of key peptidases and overall proteolytic activity typically peaking around 24 hr post blood meal (pbm) (*Barillas-Mury et al., 1991*; *Brackney et al., 2010*; *Gooding, 1966*; *Graf and Briegel, 1982*; *Graf et al., 1998*; *He et al., 2021*; *Isoe et al., 2009b*; *Noriega et al., 2002*). The transcriptional timeline is, however, more complex than a simple early translation/late transcription paradigm might suggest. Several studies have, by RT-qPCR and microarray, documented de novo transcription of peptidases and other genes peaking much earlier than 24 hr pbm (*Brackney et al., 2010*; *Isoe et al., 2009b*; *Sanders et al., 2003*). However, we found

no RNAseq data to allow us to compare gut or midgut transcriptomes genome-wide in sugar-fed conditions and across the time-course of bloodmeal digestion.

Here, we aimed to complement existing datasets by, first, contextualizing the adult female gut's transcriptome with profiles of the whole body and other major body parts and organs (head, thorax, abdomen, Malpighian tubules, ovaries) with the dual goal of (a) creating an at-a-glance reference for the anatomical distribution of transcripts, genome-wide and (b) facilitating the identification of tissue-specific marker genes which may, in the future, be useful in the construction of tissue-specific expression systems. Second, we created profiles for the five anatomically distinct regions of the adult female gut (crop, proventriculus, anterior midgut, posterior midgut, hindgut) to facilitate explorations of the regionalization of gut function. Finally, we profiled transcriptional responses to blood feeding in the anterior and posterior midgut and at multiple timepoints (6, 24, and 48 hr) after blood meal ingestion. We opted to profile *Aedes aegypti*, as it is among the most important insect vectors (*Powell, 2018*), with an increasing worldwide range (*Iwamura et al., 2020*), and a recently updated genome assembly (*Matthews et al., 2018*). All the resulting transcriptional data can be accessed at the Aegypti-Atlas online database we constructed (http://aegyptiatlas.buchonlab.com/).

With RNAseq profiles for body parts, gut regions, and blood-fed guts in hand, we launched a broad investigation into multiple aspects of mosquito biology. Our inquiries were guided by the following questions: What functions are associated with each body part? What are the functional specializations of the gut regions? How does the gut transcriptome change in the context of blood feeding, and how are those changes distributed in the midgut? What can our atlas tell us about the organization of mosquitoes' antimicrobial defenses? Finally, in a cross-species comparison between the midguts of *Ae. aegypti* and *Anopheles gambiae* (*s.l.*), we asked whether the gut's structure-function relationship is conserved between two hematophagous mosquito vector species.

## Results

### The transcriptional landscape of an adult mosquito reflects the embryonic origins and functional signatures of its major tissues

In order to develop a transcriptomic atlas of some of the main body parts of *Ae. aegypti*, we generated RNAseq profiles from the whole bodies and dissected body parts (head, thorax, abdomen, gut, Malpighian tubules, and ovaries) of sucrose-fed adult female mosquitoes of a field-derived 'Thai' strain from a region where arbovirus transmission is endemic (*League et al., 2019*). For the dissected body parts (but not whole body), we omitted the anterior-most section of the thorax as well as the final segments of the abdomen to exclude transcripts from salivary glands and from sperm in the spermatheca (see *Figure 1—figure supplement 1A* for diagram). We first evaluated the relative variance between body part and whole-body profiles by Principal Component Analysis (PCA) (*Figure 1A*). All transcriptomes from the same body part clustered together closely, providing evidence for the replicability of our data. On PC1, body parts were grouped in a manner that accorded with their divergent embryonic origin (primarily mesoderm/ectoderm for head, thorax, and abdomen, endoderm for gut and Malpighian tubules) and whole-body transcriptomes were clustered in between. Ovaries, which contain the germline, strongly separated from the other, exclusively somatic, body parts on PC2. Altogether, PCA indicated that the transcriptomes of body parts are largely defined by their embryonic origin, and that germline tissues possess a highly distinctive transcriptional signature.

Each body part we sequenced expressed a set of unique or nearly unique transcripts some of which, we hypothesize, may be exclusive tissue markers. We conducted a census of putative tissue markers (here defined as genes expressed at five or more transcripts per million (TPM) in one body part and enriched at least 50-fold compared to all other body parts). *Figure 1B* presents a selection of markers from each body part (full census in *Supplementary file 1*). The quantity of markers ranged from 8 in the abdomen to 391 in the ovaries, which expressed more markers than all other body parts combined; nearly one in fifty of all genes in the genome qualified as an ovary marker. We further noted that ovaries simultaneously expressed the greatest number of highly enriched transcripts *relative to other dissected body parts* (*Figure 1—figure supplement 1B*) and the smallest number of highly enriched transcripts *relative to whole body* (*Figure 1—figure supplement 1C*). This apparent paradox led us to hypothesize that the ovaries make a disproportionately large contribution to the transcriptome of the whole female mosquito. A gene that is uniquely expressed in the ovaries will not

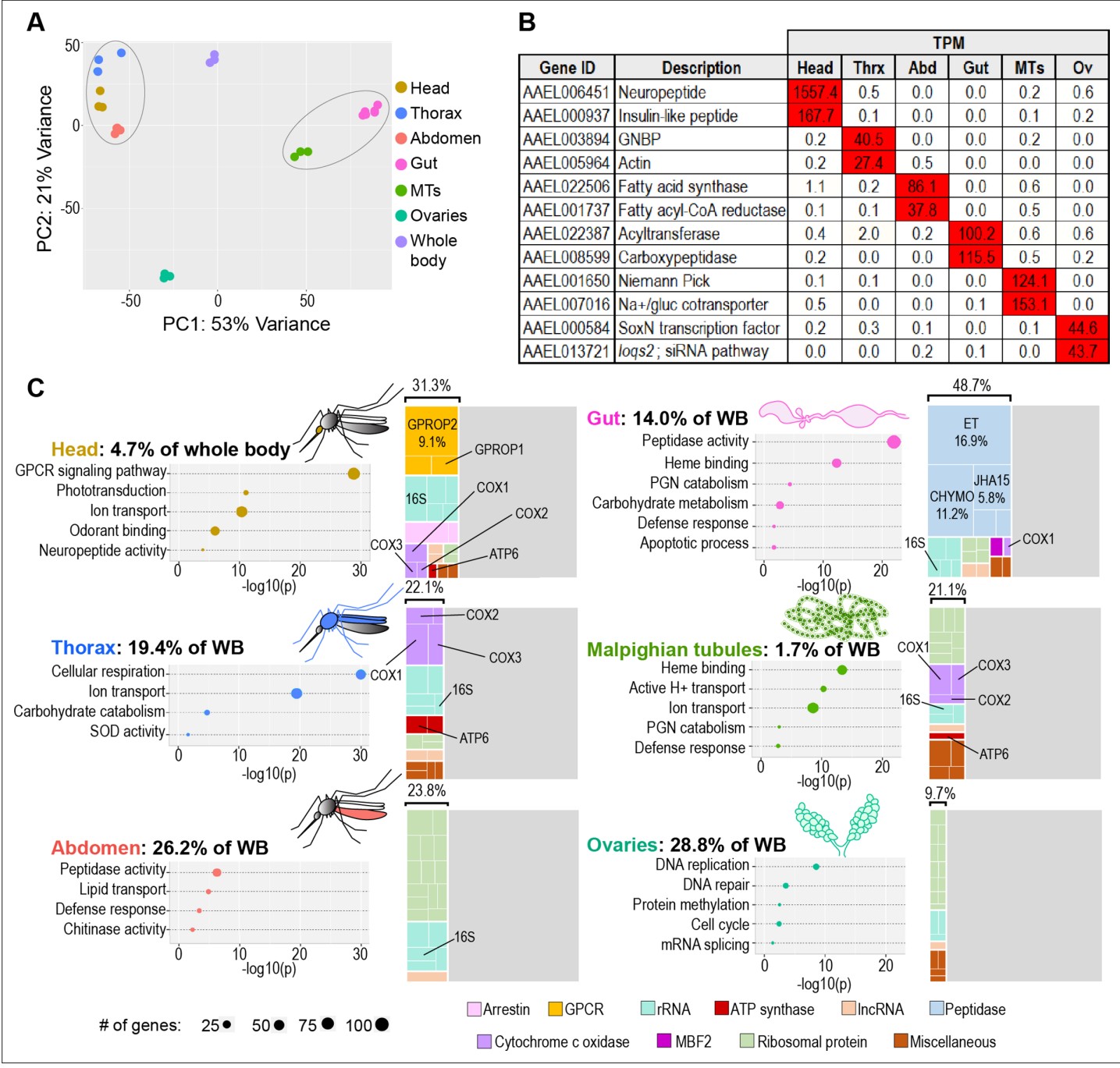

**Figure 1.** The transcriptomes of *Aedes aegypti* body parts reflect their embryonic origins and functional specializations. (**A**) Principal component analysis of the transcriptomes of whole body and body parts; 3–6 replicates per body part. Circled clusters contain body parts of dominantly endodermal (left) and mesodermal/ectodermal (right) derivation. (**B**) Expression of body-part-specific putative tissue markers (in TPM) (**C**) Calculated estimates of body parts' contributions to whole body transcriptome; Bubble plots: classic Fisher topGO gene ontology enrichment analysis of genes enriched 5 x (head, gut thorax, Malpighian tubules, abdomen) or 2 x (ovaries) relative to whole body (DESeq2, *padj* <0.05); Treemaps: 20 highest expressed genes per body part, grouped by category/function; percentages reflect the proportion of the transcriptome comprised by the displayed genes.

The online version of this article includes the following figure supplement(s) for figure 1:

**Figure supplement 1.** Ovaries possess many highly enriched genes relative to other body parts, but few relative to whole body.

**Figure supplement 2.** RT-qPCR confirms the anatomical specificity of putative marker genes.

**Figure supplement 3.** Calculated scaling factors accurately predict whole body gene expression.

be highly enriched relative to the whole mosquito body if the ovarian transcriptome comprises a large proportion of the whole body's transcriptome.

To validate our sequencing results, we performed RT-qPCR amplification of one putative marker gene from each body part. We selected a translation factor (AAEL013144) as a reference gene for our assay over more traditional candidates (*Dzaki et al., 2017*), on the basis of its robust expression and relatively small variation across samples *Figure 1—figure supplement 2A*. The results of our RT-qPCR confirmed the anatomical specificity of each selected marker (*Figure 1—figure supplement 2B* Fig).

The existence of transcripts specific to each body part allowed us to calculate their relative contribution to the whole body's transcriptome (see the 'mosquito equation' in Materials and methods). Expressed as percentages, the body part contributions we obtained were: head (4.7%), thorax (19.4%), abdomen (26.2%), gut (14.0%), Malpighian tubules (1.7%), and ovaries (28.8%). It should be noted that the percentage calculated for the thorax is likely to slightly underestimate its actual contribution, as a portion of the thorax was removed (*Figure 1—figure supplement 1A*) to eliminate salivary glands. This removal likely accounts, at least in part, for the fact that the sum of the calculated scaling factors is 94.9%, rather than 100%. We validated our estimates by using them to predict the whole-body expression of every gene in the genome, then plotting those predictions against observed values from the whole-body transcriptome (*Figure 1—figure supplement 3A*). Calculated and observed values were highly correlated (slope of 1.00, $R^2 = 0.95$) after removal of a single outlying gene (AAEL018689, a mitochondrial ribosomal RNA) which was underpredicted by our calculation. This analysis validates our hypothesis that the ovaries contribute a disproportionate number of transcripts to the whole-body transcriptome. Further, our data demonstrate that, due to the preponderance of ovarian contribution, multiple tissues have only a low representation in the body, severely limiting the ability to detect tissue-specific changes in whole-mosquito experiments.

To examine the biological processes and molecular functions enriched in each body part, we performed a Gene Ontology Enrichment Analysis (GOEA) of genes 5 x (or, for ovaries, 2 x) enriched in each body part (in comparison to whole body, DESeq2 *padj* <0.05). An extended list of GO categories may be found in *Supplementary file 2*. We complemented this approach by identifying the twenty highest expressed genes in each body part (treemaps in *Figure 1C*), reasoning that the highest expressed genes in a transcriptome may also give insight into function. In brief, the head showed enrichment for sensory machinery and neuronal signaling, the thoracic carcass (which houses flight muscle) showed signs of enhanced metabolic activity, and the abdominal carcass (which contains mostly fat body) displayed enrichment for regulatory proteolytic enzymes (CLIP-domain serine endopeptidases), defense, and lipid transport. The gut was enriched with defense-related genes including immune-activating peptidoglycan recognition proteins (PGRPs), AMPs, and immune-modulating amidase PGRPs. It also showed enrichment for digestive enzymes, especially peptidases. Notably, its three most prevalent transcripts were the non-CLIP serine endopeptidases early trypsin (*ET*), female-specific chymotrypsin (*CHYMO*), and juvenile hormone-regulated chymotrypsin (*JHA15*) which, together, accounted for more than one third of its transcriptome. (Note: Vectorbase gene IDs for all genes named in this text are listed in *Supplementary file 3*). The Malpighian tubules, which serve a function analogous to mammalian kidneys, expressed large quantities of ion transporters, especially vacuolar ATPases, which may be required to create proton gradients for use in secondary active transport (*Weng et al., 2003*). Because the ovaries did not express many highly enriched genes, we adjusted our analysis to examine genes expressed 2 x higher than in the whole body and discovered categories related to cell division and germline maintenance. We also observed that a small cohort of mitochondrial genes (*COX1*, *COX2*, *COX3*, *ATP6*, and a 16 s gene) were among the highest expressed transcripts in several body parts, especially those containing tissues that carry out energy-intensive functions (head, thorax, Malpighian tubules). Overall, we concluded that the most prominent transcripts and enriched GO categories in each body part were consistent with expectations for the tissues they house. We take these results as a validation of our data set, and as confirmation that the segments of the mosquito carcass (head, thorax, abdomen) perform well as proxies for important tissues (brain/eyes, flight muscle, fat body).

## Gut region-specific transcriptomes reveal functional compartmentalization of the mosquito midgut

To evaluate the regionalization of gut function, we generated RNAseq profiles for the five main regions of the mosquito digestive tract: crop and dorsal diverticula (hereafter referred to as 'crop' for brevity), proventriculus, anterior midgut, posterior midgut, and hindgut. A PCA of these profiles (*Figure 2A*) demonstrated clear segregation between regions, and close clustering of all replicates from the same region, indicating clearly distinct transcriptomes. The anterior and posterior midgut, which are predominantly derived from endoderm, clustered together opposite the crop and hindgut, which are of primarily ectodermal origin. The proventriculus, which is partially derived from both germ layers, was intermediate between these two clusters on PC1. Whole-gut transcriptomes clustered closely with posterior midgut transcriptomes, suggesting that the posterior midgut contributes more to the transcriptome of the whole gut than do the other regions. Accordingly, very few genes in the posterior midgut showed enrichment compared to the whole gut (*Figure 2—figure supplement 1A*). GOEA on genes enriched in each region (*Figure 2B*), complemented by an examination of the twenty highest expressed genes in each (*Figure 2C*) revealed that the ectodermally derived regions of the gut (crop and hindgut) expressed genes involved in chitin metabolism, lipid metabolism, ion transport and numerous CLIP-domain serine endopeptidases. The proventriculus was notably enriched for wingless signaling and defensive genes (lysozymes, and the highly expressed AMP gambicin, *GAM1*). In the anterior midgut, carbohydrate metabolism and heme-binding proteins (predominantly cytochrome P450s) were the most prominently enriched categories. The highest expressed gene in this region was a member of the MBF2 family of transcription factors of unknown function in mosquitoes. As the posterior midgut transcriptome closely resembled that of the whole gut, only a handful of genes were enriched 5 x in this region relative to whole gut. We also noted that a large proportion of the genome was either unexpressed or little-expressed in the posterior midgut relative to the other gut regions (*Figure 2—figure supplement 1B*). To better identify genes enriched in the posterior midgut, we calculated enrichment against an aggregate of the other four gut regions. We found that the posterior midgut is highly enriched for peptidases (especially non-CLIP serine endopeptidases) and genes involved in translation. Other enriched categories include signal recognition particle (SRP)-dependent targeting, amino acid transport and amino acid metabolism. Together these categories describe a region that is primed to translate and secrete large quantities of peptidases, and to process and absorb the resulting free amino acids. Overall, our data demonstrate that the five gut regions are highly specialized functional units.

The central function of the gut is digestion, a highly sequential process. To better understand how digestion and nutrient absorption are distributed along the gut, we examined the regional cumulative expression of enzymes and transporters putatively involved in the digestion and absorption of lipids, carbohydrates, and proteins/amino acids. We postulated that the cumulative expression of a given set of digestive enzymes/transporters of common function reflects the *investment* of each region in that function (what fraction of their transcriptome they dedicate to that process). We first assembled lists of lipases, lipid-transporting proteins, amylases/maltases, glucosidases, sugar transporters, peptidases, and amino acid transporters with a probable digestive function (excluding genes with anticipated non-digestive function, *e.g.*, peptidases annotated as, or orthologous with, CLIP-domain serine endopeptidases, caspases, proteasome components, *etc.*). For complete lists and inclusion criteria, see *Supplementary file 4*. We summed the expression of the genes from each category (in TPM) and calculated their relative expression by region (*Figure 2D*). The highest investments in different categories of lipases were divided among several regions, with phospholipases receiving the greatest investment from crop and anterior midgut, sphingomyelinases from anterior midgut, and the remainder of lipases modestly more expressed in the posterior midgut and hindgut. The crop was the greatest investor in lipid carrier proteins and fatty acid transporters, while the anterior midgut showed the greatest investment in phospholipid and sterol transporters. The highest investments in sugar digestion/absorption and protein digestion were found in the anterior and posterior midgut, respectively. The hindgut was a disproportionate investor in amino acid transporters, suggesting that many of the products of protein digestion from the midgut are absorbed there. These data demonstrate that distinct gut regions invest different amounts of their respective transcriptomes in the digestion and absorption of different nutrients, hinting at a system of specialization/sequential processing of ingested materials.

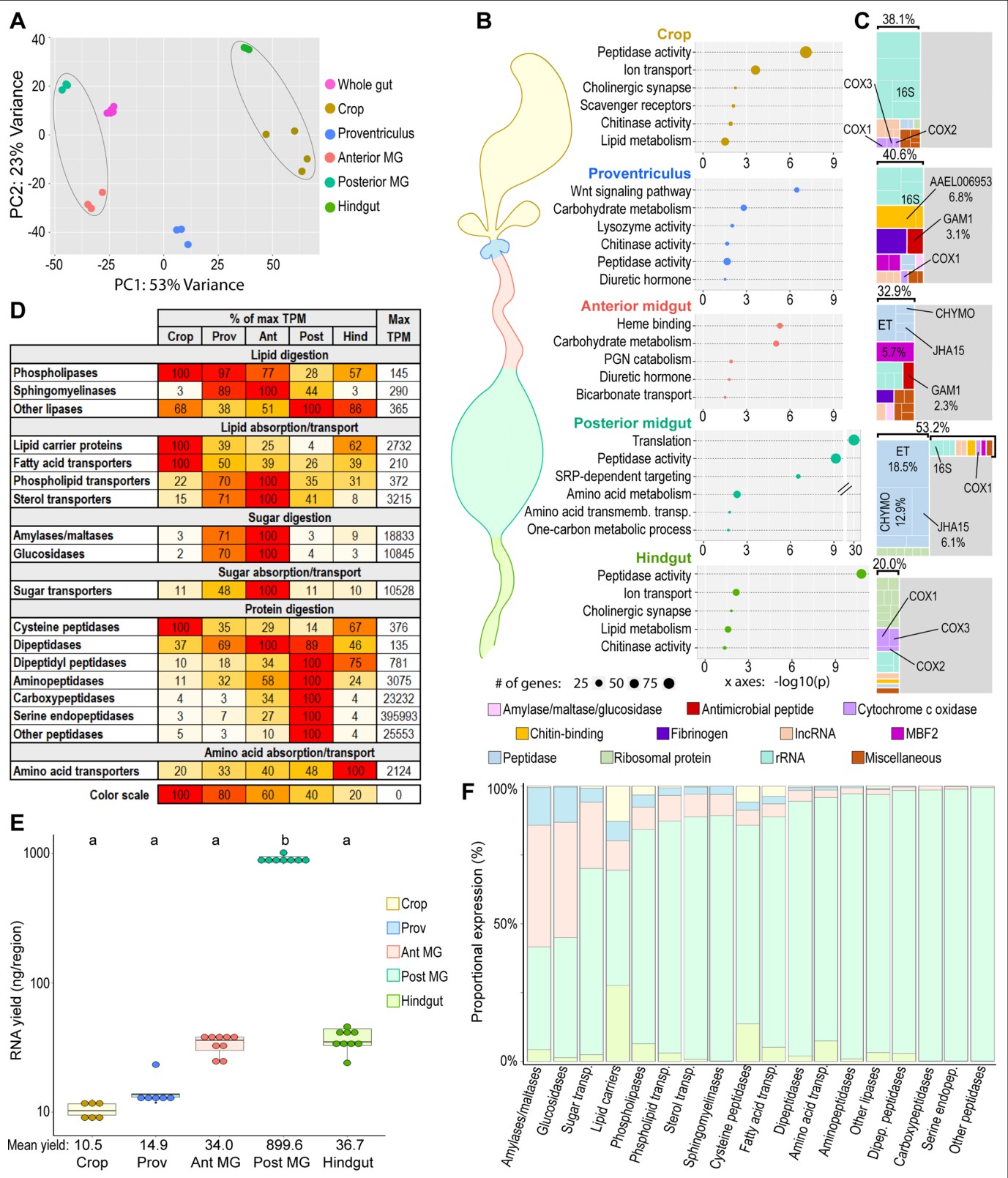

**Figure 2.** Mosquito gut regions' transcriptomic investments and outputs reveal digestive specializations. (**A**) Principal component analysis of the transcriptomes of whole gut and gut regions; 3–6 replicates. Circled clusters contain body parts of partial endodermal (left) and ectodermal (right) derivation. (**B**) Bubble plots: classic Fisher topGO gene ontology enrichment analysis of regionally enriched genes (for crop, proventriculus, and anterior midgut: 5 x enriched over whole-gut, DESeq2 padj < 0.05; for posterior midgut: enriched over all other gut regions, DESeq2 padj < 0.05) (**C**) Treemaps:

*Figure 2 continued on next page*

*Figure 2 continued*

20 highest expressed genes per region, grouped by category/function; percentages reflect the proportion of the transcriptome comprised by the displayed genes. (**D**) Regional investments in categories of digestive enzymes and transporters. Colored cells are scaled as percentages of the region with the highest transcriptional investment in the category (as measured by cumulative TPM). The region with the highest investment is scored as '100', with its cumulative TPM displayed in the rightmost column ('Max TPM'). (**E**) Estimates of regions' transcriptional output by empirical quantification of RNA yield; statistics: One-way ANOVA, Tukey HSD. (**F**) Estimated output of digestive enzymes/transporters by region (categorical investments weighted by regional RNA yield).

The online version of this article includes the following figure supplement(s) for figure 2:

**Figure supplement 1.** The transcriptome of the posterior midgut displays little deviation from the whole-gut transcriptome and low transcriptomic diversity.

Caution should be used in the interpretation of cumulative expression data, as *investment* (in TPM) is distinct from *output* (i.e. the number of transcripts of a given gene or category produced in a region). The five regions do not produce equal numbers of transcripts overall, so their transcriptional investments must be scaled by their total transcriptional *yield* to achieve an estimate of their relative contributions to specific functions. We measured the RNA content from each region (*Figure 2E*) and scaled the cumulative expression values from *Figure 2D* by yield to gain an estimate of the transcriptional output of each region with respect to each digestive category (*Figure 2F*). We found that the posterior midgut, by virtue of its outsized transcriptional yield, is the dominant contributor not only of peptidases, but of most categories of transcripts with digestive and absorptive functions. However, the proventriculus and anterior midgut together are the source of more than half of all amylases/maltases and glucosidases and contribute a sizeable minority of all sugar transporters. These observations cement our conclusion that the anterior midgut specializes in the digestion and absorption of carbohydrates, while the posterior midgut is responsible for most protein and lipid digestion.

## Blood feeding initiates a series of transcriptomic shifts over the course of digestion

The gut of the hematophagous mosquito undergoes dramatic changes upon blood feeding. To better understand these changes at a transcriptome-wide level, we generated RNAseq profiles for sugar-fed guts as well as guts at 4–6 hr (hereafter referred to as 6 hr for brevity), 24 hr, and 48 hr after feeding on a live chicken. PCA (*Figure 3A*) demonstrates that the transcriptome changes throughout at least the first 24 hr pbm, before reverting to a near-basal condition by 48 hr pbm, suggesting that the transcriptional regulation of digestion is a very dynamic process.

To evaluate the functional changes to the blood-fed gut's transcriptome, we performed a GOEA (*Figure 3B*) comparing the gut at each time-point to its state at the time-point prior. Guts at the final timepoint (48 hr) were also compared to sugar-fed guts. The analysis was limited to genes that were up or downregulated at least two-fold (DESeq2 *padj* <0.05).

We found that, at 24 hr pbm, after ecdysone titers reach their peak, (*Hagedorn et al., 1975*), steroid receptor activity was significantly enriched. A total of five steroid receptor genes were upregulated, including *EcR*, *USP* (encoding the co-receptor to EcR), *Hnf4*, *ftz-f1*, and *HR3*. Multiple categories pertaining to protein synthesis and secretion were also regulated by blood-feeding: at 6 hr pbm, a small number of genes associated with mRNA splicing, protein folding, and SRP-dependent secretion were upregulated, while genes associated with miRNA inhibition of translation were downregulated. However, by 24 hr pbm, all these changes were reversed and, additionally, genes associated with peptide transport (39 genes) and translation (57 genes) were significantly downregulated. At 48 hr pbm, protein folding, peptide transport, and translation were all upregulated again. Notably, translation was one of very few categories that was significantly enriched in 48 hr blood-fed guts compared to exclusively sugar-fed guts.

Several of the categories regulated by blood feeding appear to reflect necessary adaptations to the changing luminal environment. Six hrs following blood meal ingestion, ferric iron-binding proteins (including three ferritin subunit precursors, AAEL004335, AAEL007383, and AAEL007385) were upregulated and two genes in the lysozyme activity category (*LYSC11*, and *LYSC4*) were also induced, possibly in response to the previously described increase of ROS and bacterial density pbm (*Gusmão et al., 2010*). At 24 hr pbm, the defense response category was also induced. Two defensins (*DEFA* and *DEFD*) were enriched, as well as two peptidoglycan-degrading amidases (*PGRP-SC1*, and

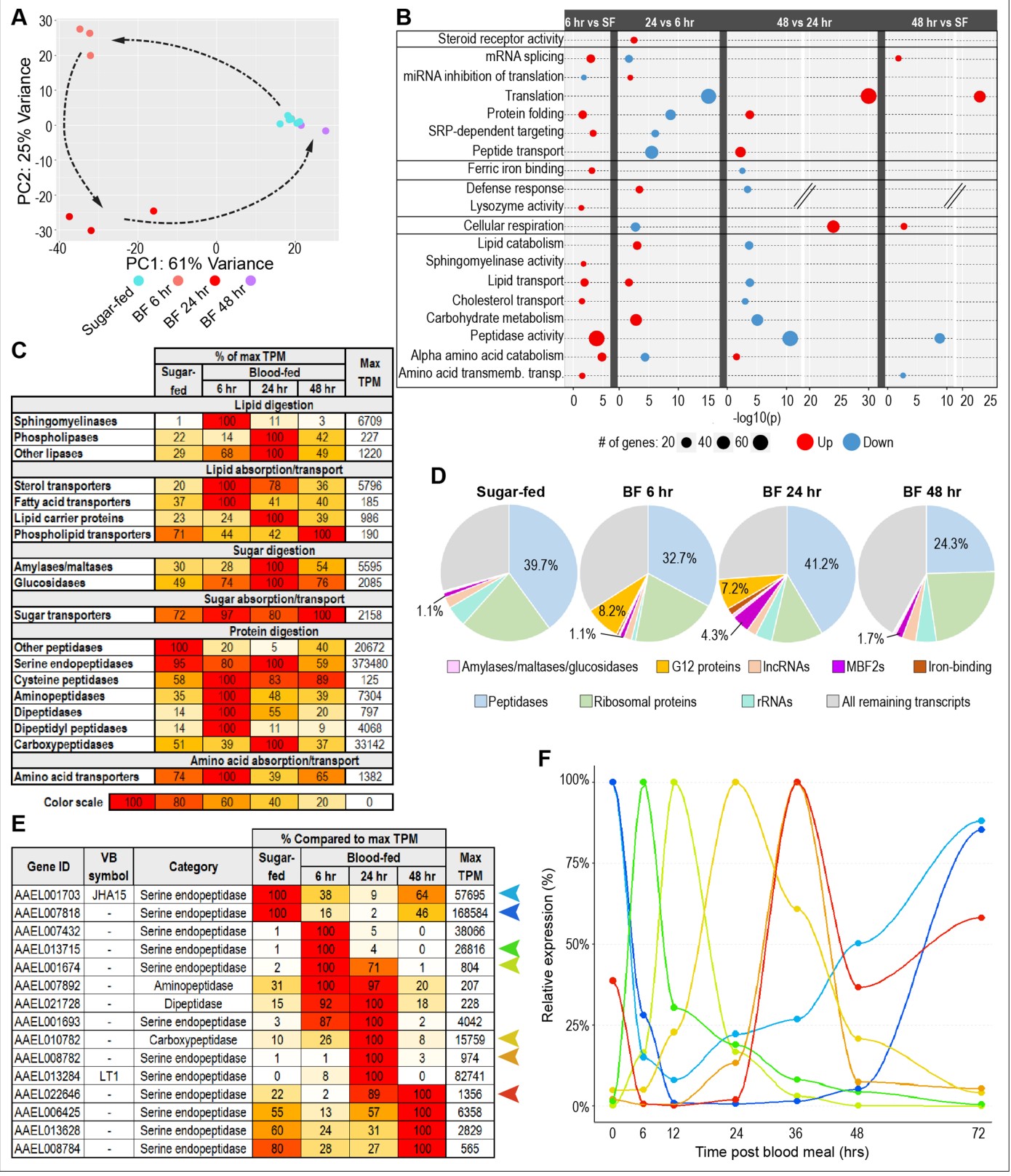

**Figure 3.** The gut's transcriptome shifts continuously throughout blood meal digestion. (**A**) Principal component analysis of the transcriptomes of sugar-fed and blood-fed guts; 3–6 replicates per condition. (**B**) Classic Fisher topGO gene ontology enrichment analysis of blood-fed guts at each timepoint pbm, in comparison to prior timepoint (DESeq2, *padj* <0.05, minimum of 2 x fold-change). (**C**) Investments in categories of digestive enzymes and transporters by timepoint/condition. Colored cells are scaled as percentages of the region with the highest transcriptional investment in the category

*Figure 3 continued on next page*

*Figure 3 continued*

(as measured by cumulative TPM). The region with the highest investment is scored as '100', with its cumulative TPM displayed in the rightmost column. (**D**) Proportional investment (out of whole transcriptome) in selected gene categories/families by timepoint/condition. (**E**) Expression of selected peptidases, scaled as a percentage of maximum expression. The region with the highest expression (in TPM) is scored as '100', with its TPM displayed in the rightmost column. (**F**) Relative expression of selected peptidases at 0, 6, 12, 24, 36, 48, and 72 hrs post-blood meal measured by RT-qPCR; reference gene: AAEL009653; 3 replicates.

The online version of this article includes the following figure supplement(s) for figure 3:

**Figure supplement 1.** Increasing RNA yield drives higher gut peptidase output at 24 hr post blood meal.

**Figure supplement 2.** Gut peptidase transcripts are induced and depleted asynchronously during blood meal digestion.

**Figure supplement 3.** Temporal clustering of serine endopeptidases is independent of physical proximity and sequence similarity.

*PGRPS4*), orthologs of which, in *Drosophila*, modulate the activation of the immune deficiency (IMD) pathway (**Buchon et al., 2009**). At 48 hr pbm, both the ferric iron-binding and defense response categories were downregulated relative to 24 hr, which is unsurprising as most of the luminal content had been evacuated by that time (**Figure 3—figure supplement 1A**). We also noted a significant induction of genes involved in cellular respiration which could reflect the metabolic expenditure required for the peristaltic evacuation of the blood bolus. GO categories related to the digestion and absorption of macronutrients were nearly all upregulated at 6 hr pbm, with some upregulated again at 24 hr but downregulated to return to baseline by 48 hr. In summary, GOEA indicates that blood feeding regulates the gut's transcription of genes involved in diverse functions, including digestion, absorption, translation, and defense.

In adult mosquitoes, a peritrophic matrix is synthesized in the hours immediately following the ingestion of a blood meal. Previous studies have documented or proposed the involvement of several genes in peritrophic matrix synthesis in *Ae. aegypti*. These include the glucosamine-fructose-6-phosphate amino transferase *AeGfat-1*, which catalyzes the first step of chitin biosynthesis, the chitin synthase *AeCs*, which catalyzes the final step (**Kato et al., 2006**), and the peritrophins AAEL004798, AAEL006953, and *Aper50* (**Shao et al., 2005**; **Whiten et al., 2018**). All five of these genes were robustly expressed in our blood-fed guts, and two (*AeGfat-1* and *Aper50*) were significantly upregulated at 6 hr pbm. Notably, the peritrophin AAEL006953 was heavily concentrated in the proventriculus at baseline and, at a TPM of over 68,000, was that region's top expressed gene (**Figure 2B**). Other candidates, orthologous to *Drosophila* chitin synthases (AAEL002718) and peritrophins (AAEL005702, AAEL009585) showed little expression in the gut at any timepoint. Surprisingly, *DUOX*, the ortholog of a dual oxidase which catalyzes the formation of a dityrosine network in the mucin layer adjacent to the *An. gambia* peritrophic matrix (**Kumar et al., 2010**), was notably absent in the *Ae. aegypti* gut (**Figure 3—figure supplement 1B**).

To characterize the blood-fed gut's changes in digestion/absorption in more detail, we calculated the cumulative investment of the gut in each digestive category at each timepoint (**Figure 3C**). Overall, cumulative transcript analysis echoed the results of our GOEA. Both showed that the gut's transcriptional investment in sphingomyelinases and amino acid transporters peaked at 6 hr pbm, and that the expression of carbohydrate digestive enzymes was elevated at 24 hr pbm but declined again at 48 hr pbm. Our granular cumulative expression analysis revealed that aminopeptidases, dipeptidases, and dipeptidyl peptidases reached peak investment at 6 hr, and carboxypeptidases at 24 hr. Remarkably, the gut's investment in serine endopeptidases (non-CLIP), which constitute the majority of peptidase transcripts in the gut by nearly an order of magnitude, remained steady across sugar-fed, 6 hr, and 24 hr pbm guts, before plunging by approximately one third at 48 hr pbm.

We extended our cumulative investment approach to other functions in the gut by aggregating transcripts of other transcriptionally prominent families and comparing their relative proportions across all timepoints (**Figure 3D**). This revealed the dramatic temporary increase in the expression of a family of twelve genes containing the insect allergen domain (IPR010629) which are orthologous to the *An. gambiae* G12 gene and have been found to possess hemolytic, cytolytic, and antiviral properties (**Foo et al., 2021**). We also observed that the family of MBF2 transcription factors - one member of which was previously noted among the highest expressed genes of the anterior midgut (**Figure 2C**) - surged from just over 1% of the transcriptome to more than 4% at 24 hr pbm, partially subsiding to approximately 1.7% by 48 hr pbm. Our comparison of summed transcripts yielded an

apparent paradox: while GOEA demonstrated the upregulation of multiple peptidases at 6 hr pbm, the gut's overall investment in peptidases (cumulative TPM) had actually declined at this timepoint from 39.7% to 32.7% of the transcriptome, before rising to 41.2% at 24 hr pbm. This contradiction is explained by the depletion of transcripts from the small 'early' cohort of peptidases (*ET*, *CHYMO*, and *JHA15*) which compensated for the upregulation of other peptidases at 6 hr pbm. It should be noted that, while peptidase *investment* did not increase much in blood-fed guts, higher RNA yield *Figure 3—figure supplement 1C* caused peptidase transcript *output* to increase significantly at 24 hr pbm (*Figure 3—figure supplement 1D*).

To more closely examine the dynamics of peptidase transcription in the blood-fed gut, we performed a clustering analysis (*Figure 3—figure supplement 2A*) which revealed that the transcript levels of many peptidases in the gut change dramatically over the course of blood meal digestion, that these changes come in the form of both induction and repression/depletion, frequently followed by a return to baseline at the subsequent timepoint, and that they manifest in multiple successive waves. Our clustering analysis divided the dynamically expressed gut peptidases into three waves of induced transcripts (rapid, intermediate, and delayed), and three corresponding waves of repressed or depleted transcripts. Among induced peptidases, the rapid and delayed waves comprise genes that peaked in our data set at 6 hr and 24 hr pbm, (23 and 40 genes respectively). Our clustering analysis implied that another, 'intermediate' cohort of induced peptidases (16 genes) nests somewhere between these waves. These genes show sustained induction from 6 to 24 hr, and likely peak at some time between these two timepoints. The waves of rapid, intermediate, and delayed repressed/depleted peptidases mirror the expression patterns of their induced counterparts in inverse. *Figure 3E* shows the expression of representative peptidases from each of the six cohorts across the blood feeding kinetic. We validated our findings by performing RT-qPCR on selected peptidases (two repressed/depleted and five induced genes) at 6, 12, 24, 36, 48, and 72 hr pbm. We selected *RpS30* (AAEL009653), a housekeeping gene with robust and steady expression over the course of blood meal digestion, as a reference gene for our assay (*Figure 3—figure supplement 2B*). *Figure 3F* shows the relative expression of each peptidase (for unscaled data normalized only to the reference gene, see *Figure 3—figure supplement 2C*). At each timepoint we assayed, we captured one or more genes peaking sharply, further illustrating the dynamic nature of peptidase expression in the blood-fed gut. Most peptidases are organized in genomic clusters which, we hypothesized, could underlie co-regulation. We also examined the evolutionary relatedness of the peptidases from each temporal cluster by comparing their positions on a phylogenetic tree (composed using Geneious R11 software, *Figure 3—figure supplement 3A*). While sequence similarity corresponded closely with physical location in most cases, neither parameter appeared correlated with temporal clustering. Altogether, our data reveal that blood feeding initiates a well-orchestrated transcriptomic response in the gut, dominated by the dynamic up and down-regulation of peptidases throughout the entire process of digestion.

## Regional digestive specializations are preserved in the blood-fed midgut

Considering the extensive functional regionalization of the *Ae. aegypti* gut and the profound reshaping of the whole-gut transcriptome upon blood feeding, we asked whether regional differences are preserved in the blood-fed gut. To evaluate the regionalized effects of blood feeding, we generated transcriptomes for dissected anterior midguts (without proventriculus) and posterior midguts at 24 hr pbm. PCA (*Figure 4A*) revealed that all profiles clustered both by region and by blood feeding status. As we previously noted (*Figure 2A*), there was little difference between sugar-fed whole gut and posterior midgut profiles. Upon blood feeding, both whole gut and posterior midgut shifted in tandem, maintaining proximity, indicating that blood feeding does not lessen the dominance of the posterior midgut's transcriptional yield over that of the anterior. Both midgut regions shifted in response to blood feeding; moreover, they moved in the same directions with respect to PC1 and PC2, suggesting that the overall transcriptional responses of the two regions share some commonality. We noted that the relative investments of these two regions in most of the prominent functional categories did not change dramatically upon blood feeding (*Figure 4B*). However, two of the most remarkable changes in the transcriptome of blood-fed mosquitoes, the upregulation of MBF2 transcription factors and G12 genes (*Figure 3D*), were mainly localized to the posterior midgut.

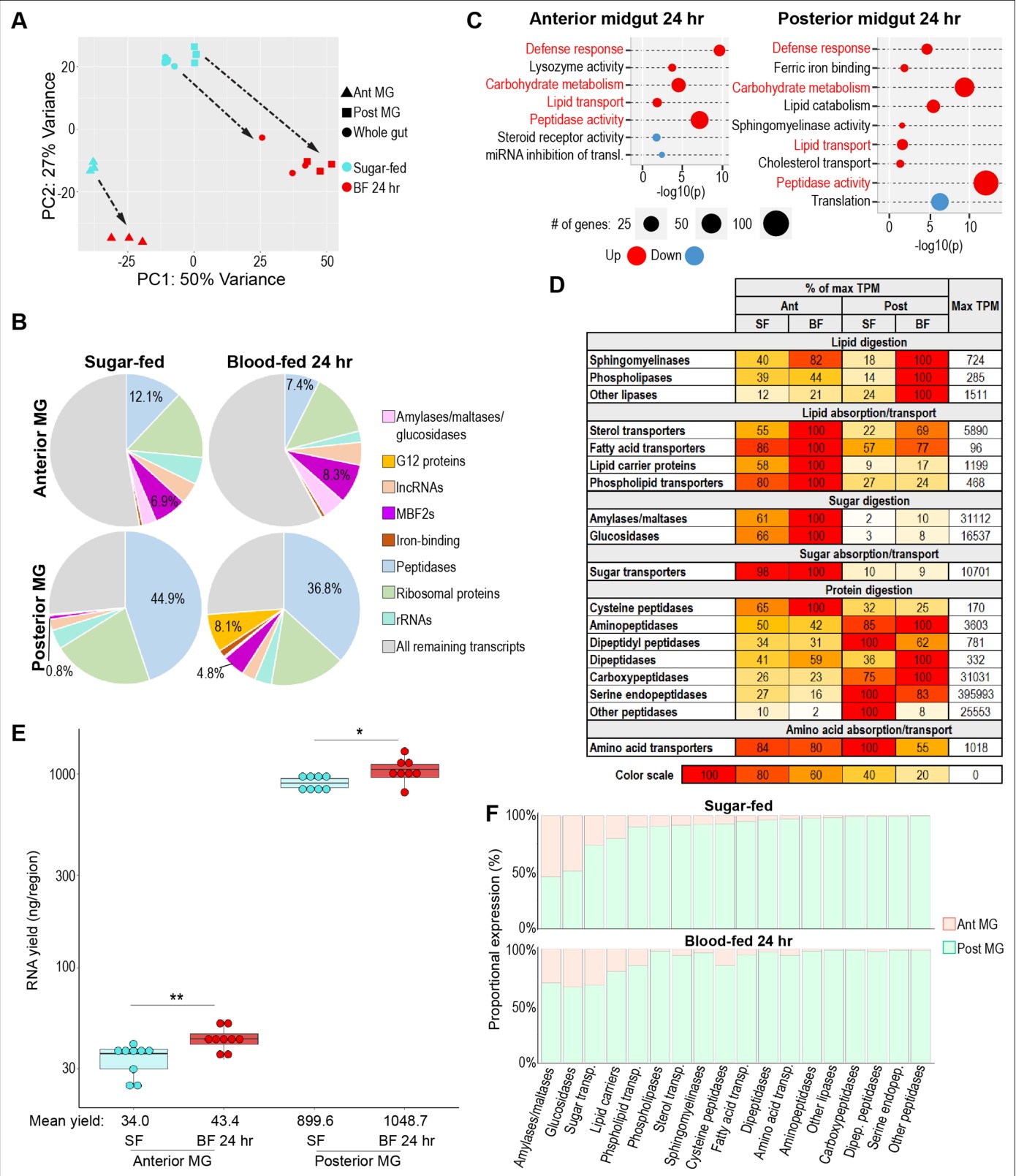

**Figure 4.** Regional specializations in sugar and protein digestion are preserved in the blood-fed midgut. (**A**) Principal component analysis of the transcriptomes of sugar-fed and 24 hr blood-fed whole guts and anterior and posterior midguts; 3–6 replicates per condition. (**B**) Proportional expression (out of whole transcriptome) of selected gene categories/families by region/condition. (**C**) Classic Fisher topGO gene ontology enrichment analysis of blood-fed versus sugar-fed regions (DESeq2, *padj* <0.05, minimum of 2 × fold-change). Categories that are regulated in both regions are

*Figure 4 continued on next page*

*Figure 4 continued*

shown in red. (**D**) Investments in categories of digestive enzymes and transporters by region/condition. Colored cells are scaled as percentages of the region/condition with the highest transcriptional investment in the category (as measured by cumulative TPM). The treatment with the highest investment is scored as '100', with its cumulative TPM displayed in the rightmost column. (**E**) Estimates of midgut regions' transcriptional yield by empirical quantification of RNA in sugar-fed guts versus 24 hr after feeding with an artificial blood replacement diet; statistics: unpaired *t*-test. (**F**) Estimated output of digestive enzymes/transporters by midgut region in sugar-fed and blood-fed mosquitoes (categorical investments weighted by regional RNA yield).

A GOEA of categories upregulated and downregulated by blood feeding in the anterior and posterior midgut (*Figure 4C*) mainly recapitulated changes already noted in blood-fed whole guts (*Figure 3B*). Most categories that were upregulated in the anterior midgut were similarly upregulated in the posterior, albeit to differing extents. The fact that blood feeding promotes enrichment of many of the same digestive/absorptive categories in both anterior and posterior midgut could imply that the two regions are converging on a common role with respect to macronutrient exploitation. However, an analysis of cumulative expression (*Figure 4D*) demonstrated that the two regions' respective investments in digestive enzymes and transporters remained highly divergent. Consistent with the results of the GOEA, the posterior midgut substantially increased its investment in lipase transcripts, with a smaller increase in the anterior midgut. By contrast, the anterior midgut kept a substantial edge in its investment in lipid transporters and lipid carrier proteins. The posterior midgut increased its investment in amylases/maltases and glucosidases by approximately five and twofold, respectively, and the anterior midgut increased both by less than twofold. However, the difference in baseline expression was such that the blood-fed anterior midgut's investment was still ten times that of the blood-fed posterior midgut. Neither region substantially increased its investment in sugar transporters, peptidases, or amino acid transporters. When these regional investments were weighted by the increased RNA yield observed in both blood-fed midgut regions (*Figure 4E*), we found that the proportional output of the posterior midgut increased across most digestive categories, but that the anterior midgut still contributed a substantial minority of transcripts for sugar-digesting enzymes and sugar transporters (*Figure 4F*). Altogether, we concluded that both midgut regions participate in the transcriptional response to blood feeding, and that the blood-fed anterior and posterior midgut retain their regional specializations with respect to carbohydrate and protein digestion.

## Immune gene patterning reveals areas of immune activity and immune tolerance in the mosquito body and gut

As mosquitoes are prolific vectors of important human pathogens, their defensive functions are of paramount interest. We assembled a table recording the expression of immune genes in each category by body part, gut region, and blood-fed timepoint (*Supplementary file 5*). Many genes associated with the siRNA and piRNA pathways display a strong tropism toward the ovaries (*Figure 5A*), possibly indicating that, in mosquitoes, siRNAs as well as piRNAs (*Siomi et al., 2011*) are required to protect the germline from selfish elements. Accordingly, the gene *loqs2*, which has been described as an essential component of *Ae. aegypti's* systemic siRNA response to Zika and dengue viruses (*Olmo et al., 2018*) was restricted to the ovaries to such an extent that it qualified as an ovary-specific marker (*Figure 1B*). The ovaries were also the site of the highest expression of genes coding for components of Toll, IMD, and JAK-STAT signaling. Remarkably, genes coding for extracellular regulators (such as most CLIP-domain serine endopeptidases) and pattern recognition proteins (PGRPs and Gram-negative bacteria-binding proteins, GNBPs) did not show this pattern of expression. Instead, they were generally highly expressed in the head, thorax and abdomen, the three compartments that contain fat body. Even more remarkably, AMPs (*Figure 5B*) showed negligible expression in the ovaries. Our results suggest that, in the female germline, portions of the IMD, Toll, and JAK-STAT pathways are uncoupled from upstream pattern recognition proteins and downstream AMP targets, possibly playing a regulatory role in some alternative process (e.g. development).

An examination of the distribution of AMPs and lysozymes in the body (*Figure 5B*) revealed that two AMPs, holotricin (*GRRP*) and gambicin, together accounted for most AMP transcripts in the mosquito body. These two AMPs were expressed in different body parts in a near mutually exclusive manner. The head, thorax, and abdomen of the mosquito predominantly expressed holotricin transcripts, with

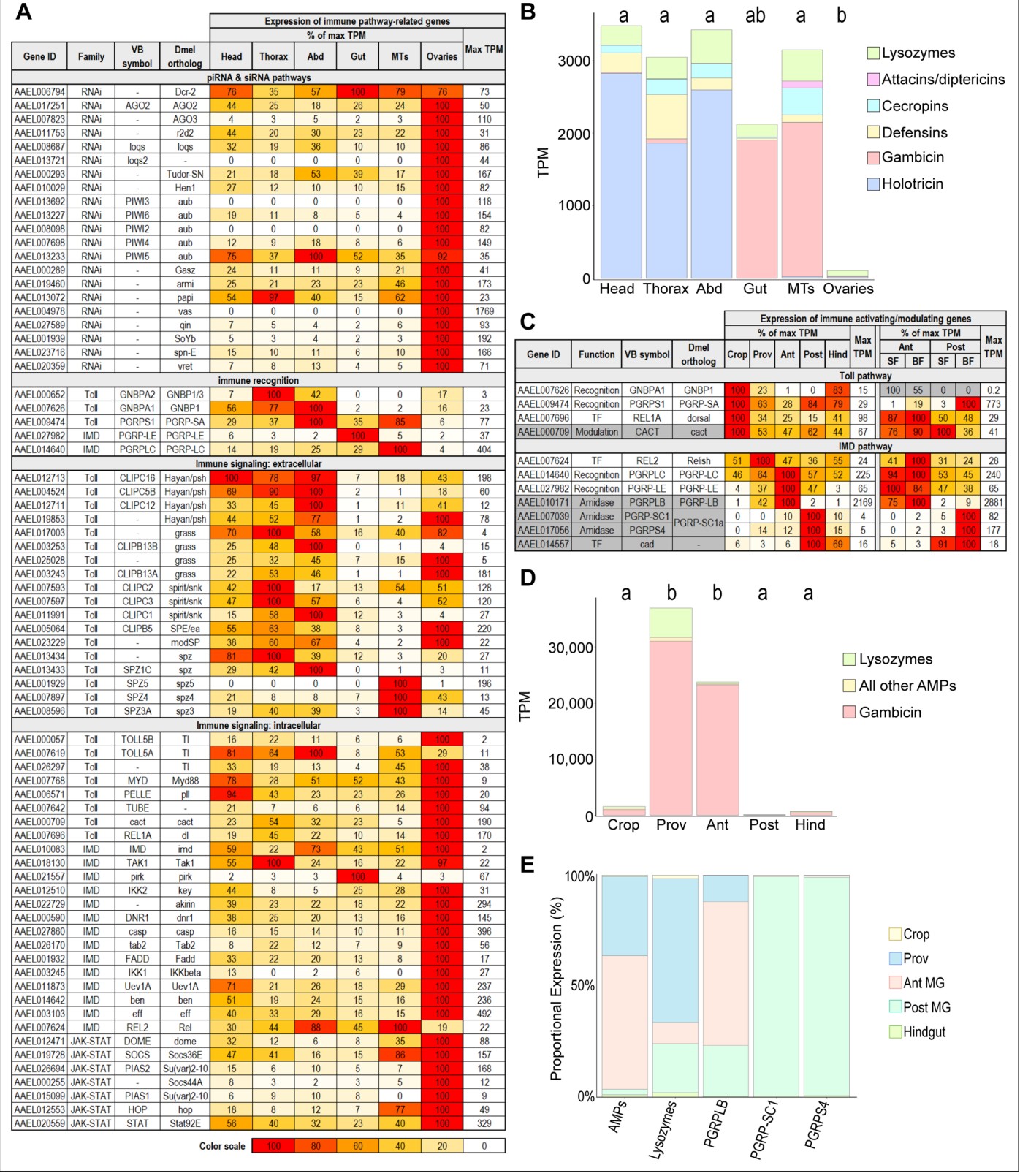

**Figure 5.** Immune gene patterning reveals zones of immune activity and tolerance in the mosquito body and gut. (**A**) Body parts' investments in immune pathway-related genes, scaled as a percentage of maximum expression. The body part with the highest expression (in TPM) is scored as '100' with its TPM displayed in the rightmost column. Note: immune-related genes may serve an alternate, developmental role in ovaries. (**B**) Cumulative antimicrobial peptide expression by body part (TPM); statistics: One-way ANOVA, Tukey HSD. (**C**) Expression of immune activating (white) /modulating

*Figure 5 continued on next page*

*Figure 5 continued*

(gray) genes, scaled as a percentage of maximum expression. The region with the highest expression (in TPM) is scored as '100' with its TPM displayed in the rightmost column of each half of the table. For genes with very low expression (Max TPM <1) color scale is replaced with gray. (**D**) Cumulative antimicrobial peptide expression by gut region (TPM); statistics: One-way ANOVA, Tukey HSD. (**E**) Output of immune-related genes and gene categories by gut region (TPM weighted by regional RNA yield).

very little gambicin, while the gut and Malpighian tubules were dominated by gambicin expression and expressed very few holotricin transcripts.

Immune activity in the gut is tightly regulated as this organ interfaces with both commensal and pathogenic microbes. In *Drosophila*, the IMD pathway is the main pathway controlling AMPs in the midgut, and its activity is strongly constrained by expression of amidase PGRPs (*Buchon et al., 2009*), while the Toll pathway is expressed mostly in the ectodermal portions of the gut. We found a similar pattern in the gut of *Ae. aegypti* (*Figure 5C*) with orthologs of Toll pathway recognition proteins most strongly expressed in the crop while IMD-activating PGRPs (*PGRPLC* and *PGRP-LE*) and IMD pathway components were enriched in the midgut. Investment in these IMD-activating PGRPs was highest in the anterior midgut, while immune-modulating PGRPs showed more divided expression. The amidase *PGRPLB* was most prominently expressed in the anterior midgut, but the short amidase PGRPs (*PGRP-SC1* and *PGRPS4*, orthologs of *Drosophila PGRP-SC1a* and *PGRP-SC1b*) were most strongly expressed in the posterior midgut. Likewise, an ortholog of caudal (*cad*), a transcription factor that limits AMP expression in the *Drosophila* midgut (*Clayton et al., 2013*; *Ryu et al., 2008*), was profoundly enriched in the posterior midgut. Altogether, we found that the expression patterns of these key genes suggested enhanced immune vigilance in the anterior portion of the midgut, with hallmarks of immune tolerance prominent in the posterior midgut. We also observed that blood feeding enhanced the expression of the short amidase PGRPs in the posterior midgut, sharpening the regional divide.

A cumulative analysis of the expression of immune effectors (AMPs and lysozymes) in the gut showed that the proventriculus and anterior midgut, respectively, invest 3.6% and 2.3% of their transcriptomes in the expression of these effectors – especially gambicin (*Figure 5D*). The posterior midgut, by contrast, expressed just 110 TPM of AMPs and lysozymes combined, supporting our hypothesis that the posterior midgut is characterized by immune tolerance toward microbes. When AMP and lysozyme expression is weighted by gut regional RNA yield (*Figure 5E*), it is apparent that the anterior midgut and proventriculus contribute the majority of transcripts in these categories despite their relatively small transcriptional yields. Overall, the expression patterns of immune activators, modulators, and effectors suggests that the anterior portions of the midgut may exert strong selection of microbes upon entry, while the posterior midgut is a region of greater immune tolerance.

## Gut digestive and defensive regionalization is well conserved between the midguts of *Aedes aegypti* and *Anopheles gambiae*

*Ae. aegypti* and *An. gambiae* are both important hematophagous vectors, and their digestive tracts bear a clear anatomical similarity. To assess how similar these organs are at the transcriptomic level, we created RNAseq profiles for whole guts (comprising all regions from crop to hindgut) and midgut regions (proventriculus, anterior midgut and posterior midgut) from *An. gambiae* (*s.l.*) G3 mosquitoes and compared them to their *Ae. aegypti* counterparts. In a clustering analysis (*Figure 6A*), we found that the profiles of midgut regions and whole guts from each species were grouped similarly, with anterior midguts intermediate between proventriculi and posterior midguts. Also, in both species, whole guts clustered between anterior and posterior midgut regions but displayed more similarity to the latter, suggesting that for *An. gambiae*, as for *Ae. aegypti*, the transcriptional yield of the posterior midgut greatly exceeds that of the anterior.

We repeated our analysis of regional transcriptomic investment in nutrient digestion and absorption by generating lists of genes encoding enzymes and transporters, in the same categories as in *Figure 2C*, summing their regional expression in TPM, and calculating their relative abundance. Since we lacked transcriptomes for whole body, crop, and hindgut for *An. gambiae*, it was necessary to adjust the inclusion criteria we employed (*Supplementary file 4*) for our previous investment analysis. Complete lists of genes in each category with adjusted inclusion criteria can be found in *Supplementary file 6* for *Ae. aegypti* and *Supplementary file 7* for *An. gambiae*. Our comparison across the

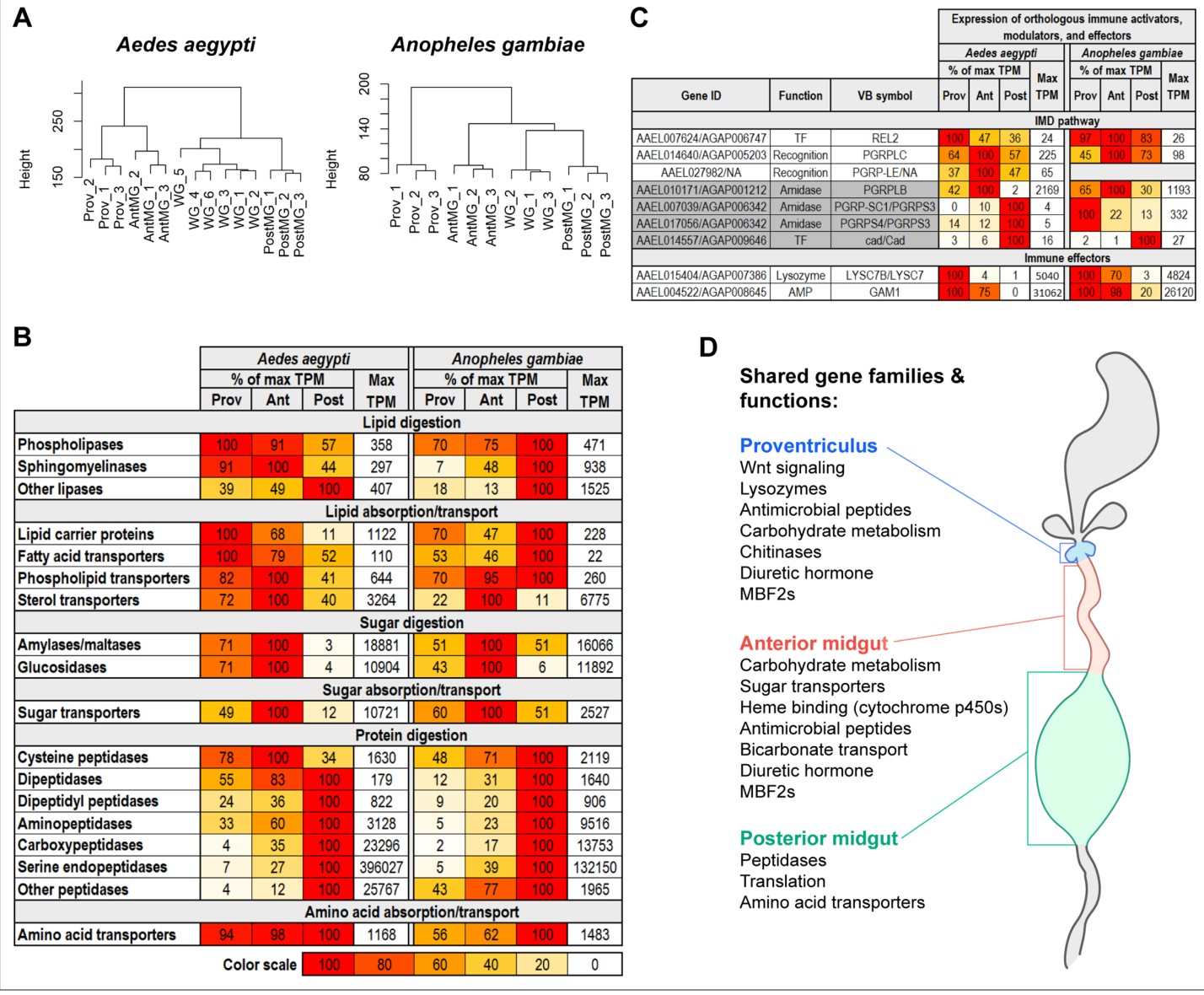

**Figure 6.** Digestive and defensive regional specializations are well conserved between the *Aedes aegypti* and *Anopheles gambiae* midguts. (**A**) Clustering analyses of the transcriptomes of *Aedes aegypti* and *Anopheles gambiae* whole gut and midgut regions; 3–6 replicates (**B**) Regional investments in categories of digestive enzymes and transporters. Colored cells are scaled as percentages of the region with the highest transcriptional investment in the category (as measured by cumulative TPM). The region with the highest investment is scored as '100', with its cumulative TPM displayed in the rightmost column of each half of the table. (**C**) Expression of immune activating (white)/modulating (gray) genes and effectors, scaled as a percentage of maximum expression. The region with the highest expression (in TPM) is scored as '100' with its TPM displayed in the rightmost column of each half of the table. (**D**) Conserved regional enrichment of gene families and functions in *Ae. aegypti* and *An. gambiae* midgut regions, as evaluated by classic Fisher topGO gene ontology enrichment analysis, investment analysis, and a comparison of the twenty highest expressed transcripts.

The online version of this article includes the following figure supplement(s) for figure 6:

**Figure supplement 1.** Midgut regions in *Aedes aegypti* and *Anopheles gambiae* display similar regional specializations, but disparate amplitudes of investment in key functions.

two species (*Figure 6B*) revealed overall conservation of gut structure-function. This was particularly evident for the anterior and posterior midguts which, respectively, remained the strongest investors in sugar and protein digestion/absorption. However, in *An. gambiae*, the posterior midgut appears to make a greater proportional investment in the digestion/absorption of both lipids and sugars. We also noted that, while the posterior midgut in both species was the dominant investor in peptidases,

the amplitude of the *Ae. aegypti* investment far outstripped that of *An. gambiae* (*Figure 6—figure supplement 1A*), particularly with respect to serine endopeptidases.

To compare the distribution of defensive functions in the two midguts, we examined the expression patterns of important orthologous immune activators, modulators, and effectors between the two mosquitoes (*Figure 6C*) and found clear similarities. In both species, the IMD-activating *PGRPLC* receptor as well as the immune-modulating *PGRPLB* amidase were most strongly expressed in the anterior midguts, the transcription factor *cad* was most strongly expressed in the posterior midguts, and the orthologs of *GAM1* and *LYSC7B* (respectively the highest expressed AMP and lysozyme in the *Ae. aegypti* midgut) were most strongly expressed in the proventriculi. This distribution of key genes leads us to hypothesize that the immune regionalization of the *An. gambiae* midgut is likely similar to that of *Ae. aegypti*. It should be noted, however, that the short amidase, *PGRPS3*, a putative negative regulator of the IMD pathway, is robustly expressed in the *An. gambiae* proventriculus, in striking contrast to its *Ae. aegypti* counterparts. We further note that, while the amplitude of *GAM1* and *LYSC7B*/*LYSC7* investment is comparable in the two species, the *An. gambaie* proventriculus and anterior midgut transcriptomes contained large quantities of transcripts for cecropins and defensins, rendering their overall AMP investment more than five times higher than the same regions in *Ae. aegypti* (*Figure 6—figure supplement 1B*). In both species, the posterior midgut's investment in AMPs was negligible. Altogether, the expression of digestive (*Figure 6B*) and defensive genes (*Figure 6C*) as well as a GOEA (*Figure 6—figure supplement 1C*) of the *An. gambiae* midgut regions confirm that the midgut structure-function relationship is well conserved between the two mosquito species. *Figure 6D* provides a summary of enriched functions and prominently expressed gene families that are shared by the midgut regions of *Ae. aegypti* and *An. gambiae*.

## A small number of disparately expressed genes differentiate *Aedes aegypti* and *Anopheles gambiae* midgut transcriptomes

The conservation of gut functional regionalization may be due either to conserved expression patterns among orthologs across evolutionary distance, or to convergent patterning of non-homologous genes of shared function. To evaluate how well the relative expression of individual orthologous genes has been maintained between *Ae. aegypti* and *An. gambiae* midgut regions, we performed a series of correlation analyses using only the 7,430 genes classified by Orthogroups Analysis (*Supplementary file 8*) as one-to-one orthologs. Scaled expression values for the orthologs were poorly correlated both in whole guts (*Figure 7A*), and in dissected regions (*Figure 7—figure supplement 1A*), with $R^2$ values ranging from 0.08 (proventriculus) to 0.56 (posterior midgut). However, the distribution of data points in the correlation graphs suggested that a few highly/disparately expressed genes substantially depressed correlation. To test the effect of these genes on overall correlation, we sequentially censored genes that were strongly and significantly disparately expressed between both species (DESeq2, *padj* <0.05), rescaled the expression (TPM) of the remaining one-to-one orthologs, and plotted the resulting changes in slope and correlation coefficient (*Figure 7B*, *Figure 7—figure supplement 1B*). For the proventriculus, anterior midgut, posterior midgut, and whole gut the censorship of, respectively 4, 10, 20, and 35 1-to-1 orthologous pairs was sufficient to reveal strong correlation between the remaining orthologs, with slopes ranging from 0.78 to 1.1 and $R^2$ values ranging from 0.77 to 0.89 (*Figure 7C*, *Figure 7—figure supplement 1C*). From this, we conclude that a 'species signature' masking the transcriptional similarity of *Ae. aegypti* and *An. gambiae* midguts is the product of a small number of highly/disparately expressed genes.

*Figure 7D* displays the genes censored from our correlation analysis in the three midgut regions. The most disparately expressed gene in both the *An. gambiae* proventriculus and anterior midgut encodes an uncharacterized protein containing a domain (IPR007931) from the Tsetse EP protein which, in tsetse flies has been shown to play a role in resisting the establishment of trypanosome infections (*Haines et al., 2010*). MBF2 transcription factors were also censored in each region: twice they were expressed higher in *Ae. aegypti* (proventriculus and anterior midgut) and once in *An. gambiae* (posterior midgut). Among the disparately expressed orthologs, the most abundant category was ribosomal proteins. We observed that in both the anterior and posterior midgut, *Ae. aegypti* had significantly increased the expression of some ribosomal proteins relative to their one-to-one orthologs in *An. gambiae*, and *vice versa*. Phylogenetic analysis of ribosomal proteins (*Figure 7—figure supplement 2A*) confirmed that the putatively one-to-one orthologous ribosomal genes identified

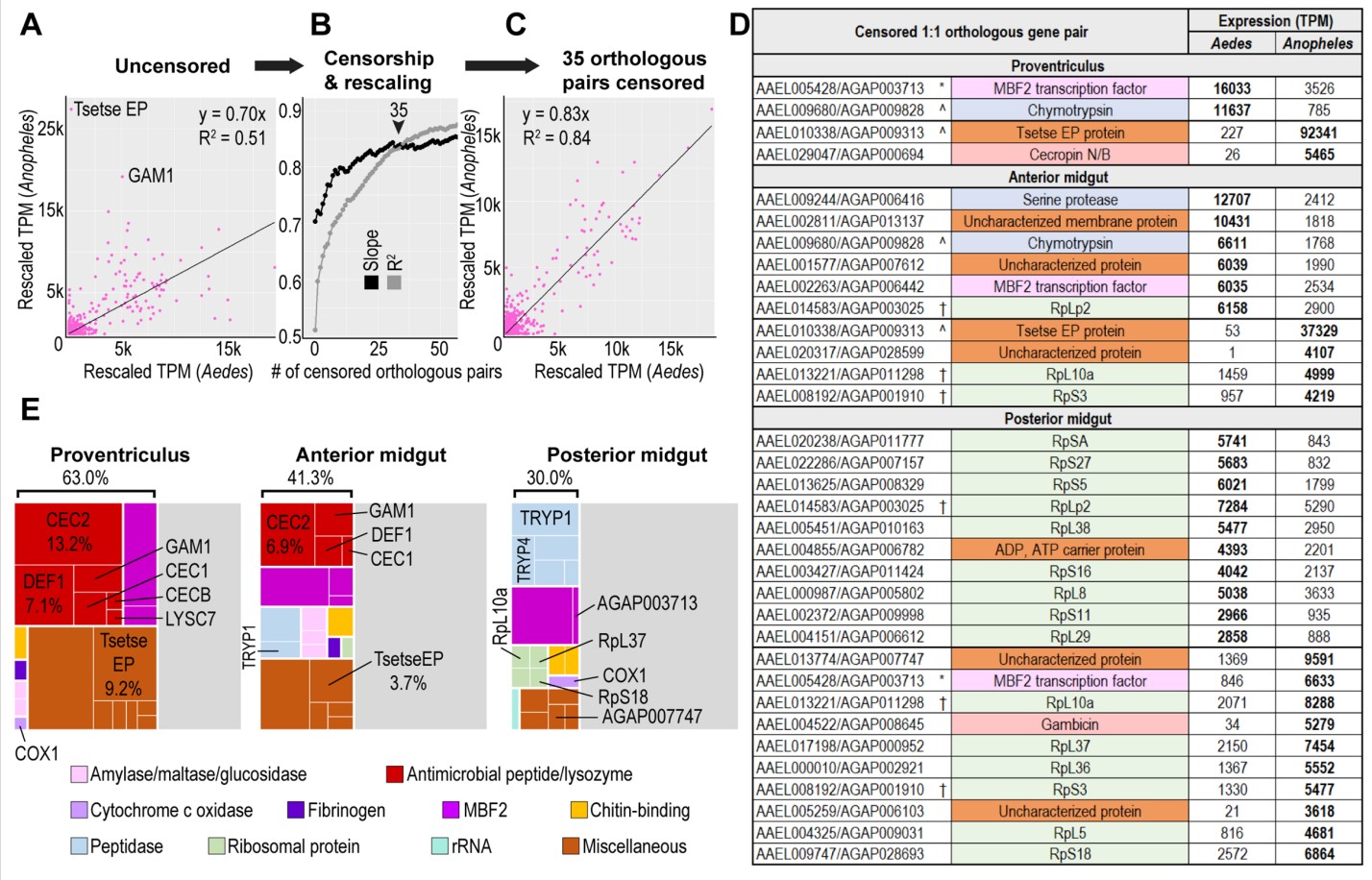

**Figure 7.** A small number of disparately expressed genes constitute a 'species signature' differentiating midgut transcriptomes in *Aedes aegypti* and *Anopheles gambiae*. (**A–C**) Correlation of one-to-one orthologous genes before and after censorship of the 35 most disparately expressed orthologous pairs. (**D**) One-to-one orthologs censored in a regional correlation analysis; (^) signifies orthologous pairs censored in both proventriculus and anterior midgut, (†) signifies orthologous pairs censored in both anterior midgut and posterior midgut (*) signifies orthologous pairs censored in both proventriculus and posterior midgut. (**E**) 20 highest expressed transcripts per region, grouped by category/function; percentages reflect the proportion of the transcriptome comprised by the displayed genes.

The online version of this article includes the following figure supplement(s) for figure 7:

**Figure supplement 1.** A small number of highly/disparately transcribed genes obscure the transcriptional correlation between one-to-one orthologous genes in *Aedes aegypti* and *Anopheles gambiae* midgut regions.

**Figure supplement 2.** Phylogenetic analysis confirms disparately expressed one-to-one orthologous ribosomal proteins are sister sequences.

**Figure supplement 3.** A model for regionalized microbe selection and immune tolerance in the mosquito gut.

by Orthogroups Analysis were sister sequences and that the disparately expressed pairs of orthologous genes were widely distributed throughout the phylogenetic tree. We therefore conclude that evolutionary changes between the two species have reshuffled the expression of ribosomal proteins, significantly reducing the roles of some while increasing those of others.

Many of the genes censored in our correlation analysis were among the highest expressed genes in the three *Ae. aegypti* and *An. gambiae* midgut regions. A follow-up comparison of the 20 highest expressed genes in the midgut regions of the two species (*Figures 2C and 7E*) revealed striking differences in both amplitude and function. In the proventriculus, *An. gambiae* devoted a higher proportion of its transcriptome (63% vs 41% in *Ae. aegypti*) to the expression of the 20 highest expressed genes. The three highest expressed, including an uncharacterized gene (AGAP013543), the Tsetse EP gene (AGAP009313), and a cecropin (*CEC2*) together composed more than one third of the transcriptome. Both species' proventriculi shared expression of two MBF2 transcription factors (AAEL005428 and AAEL013885 in *Ae. aegypti*, AGAP011630 and AGAP000570 in *An. gambiae*), as well as the AMP

gambicin (*GAM1*). However, the *An. gambiae* proventriculus also expressed three cecropins (*CEC2*, *CEC1*, *CECB*), one defensin (*DEF1*) and a lysozyme (*LYSC7*) among its top 20 genes, together totaling more than 25% of its transcriptome. Cecropins (*CEC1*, *CEC2*) and a defensin (*DEF1*) were likewise among the 20 highest expressed genes in the *An. gambiae* but not the *Ae. aegypti* anterior midgut. In the posterior midgut, the highest expressed peptidases in *An. gambiae* were much lower expressed than their counterparts in *Ae. aegypti*, and the top 20 genes in the *An. gambiae* posterior midgut included two MBF2 transcription factors (AGAP000570, AGAP003713) that were absent among the highest expressed genes in the corresponding region in *Ae. aegypti*. Altogether, we find that while gut regional specializations have been broadly conserved between *Ae. aegypti* and *An. gambiae*, and while most one-to-one orthologous genes maintain closely correlated expression patterns, this correlation is lost among a few of the highest expressed genes in the midgut, creating a 'species signature' that differentiates the two midguts' transcriptomes.

## Discussion

Aegypti-Atlas is a new resource which contextualizes and dissects the transcriptome of the sugar and blood-fed *Aedes aegypti* gut. Aegypti-Atlas will allow exploration of gene expression across the main parts/regions of the female mosquito body/gut in an easily accessible online database. In addition, we have performed a first analysis of this new dataset, examining how biological functions are distributed in the parts of the mosquito body and the regions of the gut, as well as how they change in the gut over the course of blood meal digestion. Our study demonstrates the importance of estimating investment, yield, and output in transcriptomic studies. In addition, this research has yielded new insights into such diverse areas as the regionalization of digestive function, the temporal patterning of peptidase transcription, the organization of immune defenses, and the similarities and differences between the guts of two hematophagous vectors.

### Assessments for highest expressed genes and categorical investment/output are valuable complements to GO enrichment analysis

In bioinformatic analysis of large-scale transcriptomic data sets, GOEA is often the preferred method for determining the functional role of a tissue or body part, or for detecting functional differences across conditions. We have employed GOEA to derive valuable information about the enrichment of functions in mosquito body parts, gut regions, and blood-fed conditions. However, enrichment analysis alone cannot capture all the complexities of comparative transcriptomics, as it is primarily sensitive not to quantities of transcripts from a given category, but to quantities of genes belonging to that category. Furthermore, GOEA is substantially less robust in non-model species, owing to incomplete annotations which impact many key genes. We propose that complementary approaches, such as identifying highest expressed genes and examining transcriptomic profiles through the lenses of *investment* and *output*, can help to strengthen conclusions and capture dynamics which might otherwise be overlooked. By examining the twenty highest expressed genes in each body part and gut region, we were able to identify genes that account for large proportions of the transcriptome, but which were missed by enrichment analysis (e.g. *GAM1* in the proventriculus, MBF2 factors in the anterior midgut). Comparing the top-expressed genes across transcriptomes allowed us to see that some body parts are dominated by a small number of highly expressed genes (e.g. the gut) while others spread their transcriptome more equally over the genome (e.g. the ovaries). Comparing the relative expression of one-to-one orthologs between species allowed us to identify genes that are disproportionately expressed in one species (e.g. the Tsetse EP protein, a putative antimicrobial effector in the *An. gambiae* proventriculus) and to detect a reshuffling of expression among ribosomal proteins across the two species.

Another approach we adopted – summing transcripts belonging to a functional category and comparing the sums across body parts, regions, or timepoints – allowed us to assess relative investment in the given category. We confined the use of this method to categories of genes encoding proteins that are all presumed to have similar molecular function, narrowly defined (e.g. peptidases cleaving proteins, or antimicrobial peptides killing pathogens). The utility of evaluating regional or temporal specialization by summing transcripts (investment) as opposed to counting genes (enrichment), is evident when we consider that using GOEA, the highest expressed peptidase in the gut

carries no more weight than the fifth highest – despite a difference of over 165,000 TPM, or 16.5% of the total gut transcriptome. Using this investment approach, we were also able to break large categories (e.g. peptidases) into smaller categories (e.g. carboxypeptidases, dipeptidases, etc.) for a fine-grained picture of where and when each was most expressed. There is, however, an important caveat to the interpretation of cumulative analysis as it fails to account for differential transcriptional yields between profiled body parts, regions, or conditions. We overcame this caveat by weighting transcriptional investment by RNA yield to calculate estimated regional/temporal output of digestive transcripts, from which we were able to infer how certain tasks (*e.g.*, sugar digestion) are divided between the regions of the gut, as well as how peptidase transcript output changes over the course of blood meal digestion.

## Spatial organization of gut function is conserved across mosquito species; few highly/disparately expressed genes drive transcriptomic differences

The *Ae. aegypti* gut is linearly divided into five anatomically distinct regions. With the caveat that the crop only admits sugar meals, ingested materials encounter each of these regions sequentially. The regions of the gut therefore possess an ordinal quality, corresponding to stages (either transient or prolonged) of the ingestion/digestion process. Through GOEA and quantitative evaluation of transcripts belonging to digestive categories, we discerned a clear pattern of strong investment in sugar digestion/absorption in the anterior midgut, as hypothesized by *Hecker, 1977*. Peptidases, by contrast, were predominantly expressed in the posterior midgut. These specializations were maintained under blood-fed conditions, and largely conserved in *An. gambiae* (*s.l.*) midguts. Other notable instances of conserved regional function are apparent in the proventriculus, where both species share enrichment of wingless signaling components (*Buchon et al., 2013*) and AMPS (*Hao et al., 2003*) with other dipteran species.

While the midgut regions of *Ae. aegypti* and *An. gambiae* shared many important functions, and their one-to-one orthologs maintained strikingly close transcriptional correlation, we noted that differences in the expression of a small number of highly expressed genes created large disparities in their overall categorical apportionment of transcripts. Most notably, the *An. gambiae* proventriculus and anterior midgut expressed far greater quantities of AMPs – possibly in response to some microbial presence – as well as a handful of highly expressed but poorly characterized genes, including an apparent ortholog of a tsetse midgut protein with a putative defensive function (*Haines et al., 2010*). Meanwhile, the *An. gambiae* posterior midgut expressed far fewer peptidase transcripts than its *Ae. aegypti* counterpart, and far more of an MBF2 family of transcription cofactors which, in the *Ae. aegypti* gut, was only prominent in the anterior regions of the sugar-fed midgut and, intriguingly, the blood-fed posterior midgut. In *Drosophila* and *Bombyx mori*, MBF2 factors have been shown to complex with the transcription factor FTZ-F1 and cofactor MBF1 to activate transcription of target genes (*Li et al., 1997*; *Liu et al., 2000*). We cannot say whether this role is conserved in *Ae. aegypti* and *An. gambiae*. However, the expression of MBF2 factors in both species outstripped the expression of their orthologs of *FTZ-F1* (AAEL019863, AAEL026810, AGAP005661) and *MBF1* (AAEL008768, AGAP004990) by several orders of magnitude in multiple conditions and/or gut regions (see Aegypti-Atlas website), suggesting MBF2 may not be confined to cofactoring FTZ-F1 in these species. Altogether, this analysis yielded multiple 'species signature' genes worthy of closer scrutiny and functional study.

## The anterior midgut participates in the transcriptional response to blood feeding

The anterior midgut has been held to play little or no role in the process of blood meal digestion, as the blood bolus is sealed into the posterior midgut by the formation of the peritrophic matrix shortly after ingestion (*Billingsley, 1990*). However, late in the blood meal response (24 hr) we found that the anterior midgut had increased its investment in some categories of digestive enzymes and transporters (e.g. sphingomyelinases, sterol transporters, amylases/maltases, glucosidases, *Figure 4D*) as well as its overall transcriptional yield (*Figure 4E*). While the proportional output of the anterior midgut drops relative to the posterior midgut (*Figure 4F*), it still contributes approximately one third of the transcripts for sugar-digesting enzymes in the whole midgut (exclusive of proventriculus).

This finding is congruent with Billingsley and Hecker's observation that alpha-glucosidase activity was elevated in homogenates of *Anopheles stephensi* anterior midguts at 24 hr pbm (***Billingsley and Hecker, 1991***). As the majority of amylases/maltases and glucosidases in the *Ae. aegypti* genome possess signal peptides but lack transmembrane domains, it is possible that enzymes secreted in the anterior midgut are capable of diffusing into the extra-peritrophic area in the posterior midgut where they may participate in blood meal digestion.

## Gut peptidases are dynamically up and downregulated in sequential transcriptional waves upon blood feeding

Peptidase expression in the *Ae. aegypti* gut unfolds in a series of phases, coordinated by rising and falling titers of JH and ecdysone. In the post-emergence/pre-vitellogenic phase, JH drives the transcription of a cohort of "early" peptidases, priming the gut for its first blood meal (***Bian et al., 2008***; ***Jiang et al., 1997***; ***Noriega et al., 2001***; ***Noriega et al., 1997***). Within hours of blood feeding, translation of this pool of transcripts is initiated and peptidase activity begins to increase (***Felix et al., 1991***). Next, rising ecdysone titers drive the transcription of a "late" cohort of peptidases, which complete the digestive process (***He et al., 2021***). Finally, early peptidases are transcribed anew (***Bian et al., 2008***; ***Noriega et al., 1996***) amid a postprandial surge of JH (***Hernández-Martínez et al., 2015***; ***Lucas et al., 2015***; ***Shapiro et al., 1986***; ***Zhao et al., 2016***), restoring the gut to a state of readiness for its next blood meal.

Peptidase activity peaks somewhere between 18 and 36 hr pbm (***Gooding, 1966***; ***Graf and Briegel, 1982***; ***Noriega et al., 2002***), as do transcripts for some of the highest expressed 'late' phase peptidases (***Brackney et al., 2010***; ***Isoe et al., 2009a***; ***Isoe et al., 2009b***). However, a few publications have documented that some peptidases peak at earlier timepoints (***Brackney et al., 2010***; ***Isoe et al., 2009b***; ***Sanders et al., 2003***), and that at least one of these (*SPI*) is directly responsive to/dependent on ecdysone signaling (***He et al., 2021***). Here, our genome-wide multi-timepoint series has demonstrated that these early-peaking genes are not isolated and exceptional but are, rather, members of large transcriptional cohorts. We show that peptidases in the blood-fed *Ae. aegypti* gut are expressed not only in two phases: 'early' (translational) and 'late' (transcriptional), but that the 'late' transcriptional response manifests in multiple successive waves or shifts. We have divided these into three ('rapid', 'intermediate', and 'delayed'), but because we only obtained RNAseq profiles for guts at baseline and three blood-fed timepoints, we can only speculate how many of these shifts are actually triggered over the course of blood meal digestion, and how synchronously any of the peptidases that our clustering analysis grouped together are actually regulated. Our timepoints were too broadly spaced to say at what time each gene reached its true peak, or whether peptidases that were apparently quiescent throughout blood meal digestion in fact participated in a shift that was too transient to be detected by our experimental design. Future work will likely uncover even more complexity than we have described.

The sharp temporal patterning of peptidase transcription in the blood-fed mosquito gut raises intriguing questions. How is this transcriptional choreography achieved? Is the transcription of all 'late' phase peptidases ecdysone-dependent, and are rising ecdysone levels sufficient to drive them? To what extent is the transcriptional induction of succeeding waves dependent on the activity of prior ones? Why are some 'delayed' peptidases so slow to show any transcriptional response when ecdysone titers are ascendent (e.g. AAEL008782, ***Figure 3E***)? And what mechanism effects the precipitous drop in the 'rapid' cohort of induced peptidases (ascendent at 6 hr pbm) at a timepoint (24 hr pbm) when ecdysone titers are still high (e.g. AAEL013715) (***Hagedorn et al., 1975***)? What is the adaptive significance of the rapid up and down-regulation of so many genes of shared function? We observed that the gut's investment in aminopeptidases, dipeptidases, and carboxypeptidases has distinct temporal peaks, suggesting that polypeptides may be attacked more from one terminus or the other at different times over the course of digestion. However, most of the peptidases that participate in shift-changes are endopeptidases, and it is not clear how the sequential substitution of one set of endopeptidases for another – as opposed to simultaneous expression – helps to move the process of digestion forward. We speculate that rapidly changing conditions in the blood-fed gut may alter the stability and kinetics of specific peptidases, and that different peptidases predominate at different times because they are adapted to function in specific, transient conditions obtaining at corresponding stages of the digestive process.

## The anterior and posterior midgut regions are, respectively, characterized by signs of immune activity and immune tolerance

In the midguts of both *Ae. aegypti* and *An. gambiae*, the majority of AMP transcripts are contributed by the proventriculus and anterior midgut (*Figure 5D*, *Figure 6—figure supplement 1B*) with minimal expression in the posterior midgut. Also, in both species, the transcription factor *caudal* which, in *Drosophila* (*Ryu et al., 2008*) and *An. gambiae* (*Clayton et al., 2013*) has been shown to repress AMP expression, is dominantly expressed in the posterior midgut. The expression of immune activating recognition proteins (*PGRPLC* and, in *Ae. aegypti*, *PGRP-LE*) is less profoundly patterned, but still displays some tropism toward the anterior midgut in both mosquitoes. Altogether, these patterns suggest that ingested microbes are subjected to heightened immune surveillance and antimicrobial activity in the first regions of the midgut but are thereafter well tolerated by the posterior midgut, where they may serve some mutualistic functions. We speculate that selection by gambicin in the proventriculus and anterior midgut of *Ae. aegypti* and *An. gambiae* may play an important role in determining the composition of the microbiota that seed the posterior midgut (*Figure 7—figure supplement 3A*).

## Holotricin and gambicin are highly expressed under baseline conditions in mutually exclusive regions of the body

Our examination of the expression of immune effectors yielded the observation that at baseline, AMP expression in *Ae. aegypti* is dominated by two genes: holotricin and gambicin. We observed that holotricin predominated in the carcass, while gambicin was highly expressed in the gut and Malpighian tubules. While it is not uncommon for antimicrobial effectors to display strong tropisms to specific tissues, we are not aware of comparable instances where a single AMP exhibits such overwhelming transcriptional dominance. In future work, it might be interesting to compare the activity of holotricin and gambicin against different types of microbes (Gram-positive, Gram-negative, fungi, etc.), and to examine the effects that silencing these genes has on midgut communities, and on mosquitoes' survival in the contexts of oral and systemic infection.

## Conclusion

In this manuscript we introduce Aegypti-Atlas, a repository of RNAseq data, and demonstrate how these data can yield insights into tissue function, the organization of digestive specializations, regulatory networks, and immune effectors, as well as the changes wrought by diet and evolution in mosquitoes. This resource may also be useful for the creation of functional genomic tools (e.g. tissue-specific expression systems) which will afford researchers a greater degree of control in experiments and allow for more confidence in the interpretation of results. It is our hope that Aegypti-Atlas will be valuable to other investigators in many areas of mosquito biology.

## Materials and methods

### Mosquito provenance and rearing

For all experiments, we used 5- to 12-day-old female mosquitoes of the Thai strain (*Ae. aegypti*) or the G3 strain (*An. gambiae, s.l.*) kindly provided by Laura Harrington. The Thai colony has been maintained at Cornell University since 2011 and is outcrossed annually with field-collected eggs from Thailand to relieve inbreeding and maintain field relevance. Mosquitoes were reared at a density of 200 larvae per 1 liter tray. *Ae. aegypti* received 720 mg of fish food (Hikari #04428). *An. gambiae* received 50 mg of ground fish food per day from day 1 to 4 of development, and 150 mg per day thereafter until pupation. Adults were maintained in humidified chambers (RH 75% ± 5%) at 29 °C on a diet of 10% sucrose ad libitum. Females were reared and maintained in approximately equal proportion with males for a minimum of 5 days prior to blood-feeding and/or dissection and were presumed to be mated.

### Blood feeding and mock blood feeding

For all RNA-seq experiments involving blood-fed guts, mosquitoes were starved for twenty-four hours, then blood-fed on live chickens without anesthetic in accordance with Cornell University

IACUC approved protocol #01–56. For our extended blood feeding time series by RT-qPCR, mosquitoes were fed through a membrane on rooster blood treated with the anticoagulant sodium citrate (Lampire Biological Laboratories, Pipersville, PA, catalogue #7208806). For experiments where RNA yield was estimated in guts and midgut regions under blood-fed conditions, we employed an artificial formulation (SkitoSnack) (*Gonzales et al., 2018*) fed through a membrane. This formulation was used in order to avoid any skewing of yield which might result from RNA content in a natural blood meal. For RNA yield experiments, boluses were removed during dissection.

## Dissections and RNA extraction

Mosquitoes were sacrificed by submersion in 70% ethanol and dissected in sterile PBS. For body part samples, the last 2–3 abdominal segments were removed to eliminate the sperm-containing spermatheca. Whole guts, Malpighian tubules, and ovaries were then dissected out. The residual carcass was divided into head, thorax, and abdomen. A small anterior section of the thoracic carcass containing the salivary glands was excised to exclude salivary gland transcripts from thoracic profiles. For RNAseq experiments, a minimum of 10–20 individuals per replicate was used for large body parts (e.g. whole body, abdominal carcass, whole guts) and ≥100 individuals per replicate for small gut regions (e.g. crop, proventriculus). For additional information about replicates, see *Supplementary file 9*. For RT-qPCR time-series experiments, a minimum of 10 individuals was dissected per replicate. For all experiments, a minimum of three replicates was prepared per part and/or condition.

All samples for RNAseq were homogenized in 600 µl TRIzol and stored at –80 °C. For RNAseq experiments, RNA was extracted via a modified phenol-chloroform method (*Troha et al., 2018*). In brief, samples in TRIzol were thawed and combined with 120 µl of chloroform, followed by vortexing and centrifugation for 15 min at 12,000 g, 4 °C. Between 200 and 250 µl of aqueous supernatant was removed to a fresh tube, where it was mixed first with 700 µl Buffer RLT (Qiagen RNeasy kit, cat #74004) then 500 µl 100% ethanol. This mixture was passed through a RNeasy spin column (700 µl at a time) with centrifugations of 20 s at 10,000 g, followed by a final spin for 1 min at 10,000 g to dry the column. The membrane was then washed twice with 500 µl of Buffer RPE (20 second centrifugation, 10,000 g), followed by a 1 min centrifugation at 16,000 g to dry the column. Samples were eluted in 30 µl of RNase-free water, applied directly to the membrane.

For RNA yield experiments, dissected gut regions were pooled (10 for posterior guts, 20–50 for other regions) in 1 mL Trizol, and the number in each sample was recorded. Samples were then extracted by a standard phenol-chloroform procedure. Exactly 400 µl of aqueous supernatant was recovered from each tube, and RNA was precipitated overnight with isopropanol at –20 °C. Pellets were resuspended in 20–50 µl of nuclease-free water, and concentrations were measured by Qubit. Yield was then calculated as follows: (concentration * suspension volume / # of dissected regions).

## Library preparation and sequencing

Libraries were prepared using the Quantseq 3' mRNA-seq prep kit from Lexogen according to the manufacturer's instructions. Sample quality was evaluated before and after library preparation using a fragment analyzer (Advanced Analytical). Libraries were pooled and sequenced on the Illumina Nextseq 500 platform using standard protocols for 75 bp single-end read sequencing at the Cornell Life Sciences Sequencing Core. Sequences have been deposited on NCBI (BioProject ID: PRJNA789580).

## Analysis pipelines

Quality control of raw reads was performed with fastqc (https://github.com/s-andrews/FastQC) (*Andrews et al., 2020*) and reads were trimmed by BBMap (https://jgi.doe.gov/data-and-tools/bbtools/) and then mapped to the *Ae. aegypti* transcriptome or the *An. gambiae* transcriptome *Aedes aegypti* LVP_AGWG *Aaeg*L5.2 and *Anopheles gambiae* PEST *Agam*P4.12, VectorBase, https://www.vectorbase.org/ (*Giraldo-Calderón et al., 2022*) using Salmon version 0.9.1 with default parameters. Salmon's transcript-level quantifications were then aggregated to the gene level for gene-level differential expression analysis with the "tximport" package on the R version 3.5. DEseq2 (*Love et al., 2014*) was used for differential expression analysis and PCAs were performed using custom R scripts (available upon request). GOEA was performed using the topGO package (classic Fisher method). Scripts for Salmon, topGO, Orthogroups analysis, and DESeq2 analysis, as well as DESeq2 outputs

can be accessed at GitHub (https://github.com/bingdun/RNAseq_analysis_of_Aedes_aegypti; *Bing, 2022*).

All phylogenetic trees were constructed using Geneious software version R11, with the following settings. Alignment type: Global alignment, Cost Matrix: Blosum62, Genetic Distance Model: Jukes-Cantor, Tree Build Method: Neighbor-Joining. Clustering was performed using the Pheatmap package in R with gene expression scaled by row. To calculate *z*-scores, we first censored all genes expressed at less than 2 TPM in the relevant body parts or gut regions, then employing the following expression for each gene in each body part or gut region: $z = (x - μ)/s$, where $x$ is the expression in the body part or gut region, $μ$ represents the mean expression of all body parts or gut regions, and $s$ is the standard deviation. The peptide sequences of *Aaeg*L5.2 and *Agam*P4.12 were used to find orthogroups and orthologs in the two species using OrthoFinder v2.3 (*Emms and Kelly, 2019*). The longest transcript variant of each gene was extracted to run the OrthoFinder with the following options: '-M msa -S blast -A muscle -T iqtree'.

## Calculating the 'mosquito equation'

If X is defined as the expression of a specific gene in a given body part (in TPM), and the part's initial stands for a scaling factor equal to the proportion of transcripts in the whole body derived from that part, the relationship between the expression of that gene in each body part versus in the whole body can be expressed as.

$$(X_{Head} * H) + (X_{Thorax} * T) + (X_{Abdomen} * A) + (X_{Gut} * G) + (X_{MT} * MT) + (X_{Ovaries} * O) = X_{WholeBody}$$

where $H + T + A + G + MT + O = 100\%$ Whole Body expression.

For marker genes, the expression in other body parts may be considered to be negligible, (i.e. for a head marker, $X_{Thorax}$, $X_{Abdomen}$, $X_{Gut}$, $X_{MT}$, and $X_{Ovaries}$ are all ≈ 0). Therefore.

$$X_{Head} * H ≈ X_{WholeBody}$$

We solved for the estimated scaling factor for each of our qualifying markers, then averaged the values we obtained for each body part to create an estimate of the percent of transcripts in the whole body that are contributed by each part.

## RT-qPCR

For all RT-qPCR amplifications, RNA was pretreated with DNase (Quantabio 95150–01 K) and cDNA was prepared using the qScript cDNA synthesis kit (Quantabio 95047–100) and amplified using Perfecta SYBR Green Fastmix (Quantabio 95072–012) in a Bio-Rad CFX-Connect Instrument. The primer sequences used in this study are available in *Supplementary file 10*.

## Acknowledgements

Special thanks to Jonathan Revah for creating the Aegypti-Atlas website, and to Laura Harrington, Sylvie Pitcher, and Garrett League for their kind provision of Thai and G3 strain mosquitoes. We also acknowledge members of the Buchon lab, Louise Huot, Kristin Michel, Courtney Murdock, Laura Harrington, and Jeffrey Scott for helpful comments on the manuscript.

## Additional information

### Funding

| Funder | Grant reference number | Author |
| --- | --- | --- |
| National Institute of Allergy and Infectious Diseases | 1R01AI148529-02 | Nicolas Buchon |
| National Science Foundation | IOS 2024252 | Nicolas Buchon |

| Funder | Grant reference number | Author |
|---|---|---|
| National Institute of Allergy and Infectious Diseases | 1R01AI148541-02 | Nicolas Buchon |
| National Institute on Aging | 5R21AG065733-02 | Nicolas Buchon |

The funders had no role in study design, data collection and interpretation, or the decision to submit the work for publication.

## Author contributions

Bretta Hixson, Conceptualization, Data curation, Formal analysis, Funding acquisition, Investigation, Methodology, Validation, Visualization, Writing - original draft, Writing – review and editing; Xiao-Li Bing, Xiaowei Yang, Data curation, Formal analysis, Writing – review and editing; Alessandro Bonfini, Peter Nagy, Formal analysis, Investigation; Nicolas Buchon, Conceptualization, Data curation, Formal analysis, Funding acquisition, Project administration, Resources, Supervision, Writing - original draft, Writing – review and editing

## Author ORCIDs

Bretta Hixson ⓘ http://orcid.org/0000-0001-7983-5104
Alessandro Bonfini ⓘ http://orcid.org/0000-0001-6642-8665
Peter Nagy ⓘ http://orcid.org/0000-0002-5053-0646
Nicolas Buchon ⓘ http://orcid.org/0000-0003-3636-8387

## Decision letter and Author response

Decision letter https://doi.org/10.7554/eLife.76132.sa1
Author response https://doi.org/10.7554/eLife.76132.sa2

# Additional files

## Supplementary files

• Supplementary file 1. TPMs for putative tissue markers in major body parts of *Aedes aegypti*.

• Supplementary file 2. Gene ontology categories enriched in Aedes aegypti body parts, gut regions, and blood-fed guts; and in Anopheles gambiae (s.l.) midgut regions.

• Supplementary file 3. Index of VectorBase IDs for genes named in the text.

• Supplementary file 4. TPMs for digestive enzymes and transporters expressed in the Aedes aegypti gut.

• Supplementary file 5. Relative expression (investment) and TPMs for immune-related genes in Aedes aegypti body parts, gut regions, and blood-fed guts; and in Anopheles gambiae (s.l.) midgut regions.

• Supplementary file 6. TPMs for digestive enzymes and transporters expressed in the Aedes aegypti midgut.

• Supplementary file 7. TPMs for digestive enzymes and transporters expressed in the Anopheles gambiae (s.l.) midgut.

• Supplementary file 8. Output of orthogroups analysis relating genes from Aedes aegypti and *Anopheles gambiae*.

• Supplementary file 9. Details of replicates for RNAseq experiments in *Aedes aegypti* and Anopheles gambiae (s.l.).

• Supplementary file 10. Sequences of primers for RTqPCR assays.

## Data availability

Data have been submitted to NCBI. The BioProject ID is PRJNA789580.

The following dataset was generated:

| Author(s) | Year | Dataset title | Dataset URL | Database and Identifier |
|---|---|---|---|---|
| Hixson B, Bing XL, Yang X, Bonfini A, Nagy P, Buchon N | 2022 | A transcriptomic atlas of Aedes aegypti reveals detailed functional organization of major body parts and gut regional specializations in sugar-fed and blood-fed adult females | https://www.ncbi.nlm.nih.gov/bioproject/?term=PRJNA789580 | NCBI BioProject, PRJNA789580 |

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
