## [Editor Report]

Hixson et al. provide a large overview of gene expression level of the mosquito Aedes aegypti through the use of RNA-seq. They analyse gene expression changes in the digestive tract, as well as the 3 body regions and the ovaries in various conditions. These organ-specific transcriptomes fill a hole in our understanding of mosquito vector biology and will be an excellent starting point for many researchers to produce new projects.

---

## [Decision Letter]

**Decision letter after peer review:**

Thank you for submitting your article "A transcriptomic atlas of *Aedes aegypti* reveals detailed functional organization of major body parts and gut regional specializations in sugar-fed and blood-fed adult females" for consideration by *eLife*. Your article has been reviewed by 3 peer reviewers, and the evaluation has been overseen by a Reviewing Editor and Utpal Banerjee as the Senior Editor. The following individual involved in review of your submission has agreed to reveal their identity: Emilie Pondeville (Reviewer #2).

All reviewers were quite positive on the findings reported by the manuscript and agree that the paper contains important data that will be of use to the vector biology community. There are however some issues to address (see below) An important one is the lack of clarity/detail in the Materials and methods in regards to how the libraries were constructed and details of how the comparisons across tissues were made. However, collectively they think that the paper is a valuable contribution and if the authors can clarify/revise in regards to those issues it would be a nice paper. We therefore ask you to prepare a revised submission to address the points listed below

Essential revisions:

1. As a dry/wet bioinformatician/molecular biologist, this reviewer acknowledges that this work has been beautifully executed following standard practices for mosquito rearing and following the guidelines of RNA sequencing. Though, data reproducibility is a concern in bioinformatics since different analysis pipelines may lead to dissimilar results from same datasets. While aiming to reach a broad audience of specialists and non-specialists in RNAseq analyses, this reviewer suggests that authors deposit their command lines and scripts (even if only relevant parts) in an open-access repository such as GitHub or equivalent, so others may compare their own data without technical variation. Also, raw tables with normalized gene expression (i.e., outputs of DEseq2) could also be provided for future reference, either deposited in GEO/NCBI or as tables together with the code at GitHub.

2. One major concern relates to the normalization of transcriptomic data presented here. Authors emphasize that data from some tissues such as ovaries or specific midgut regions showed considerably distinct patterns of gene expression. Consequently, such patterns of unbalanced transcriptional levels can skew normalization and could be of concern. The tool used here for normalization of transcript counts, DEseq2, should not be severely impacted by these events since, as mentioned by its author Michael Love, "the median ratio normalization in DESeq2 doesn't have as strong of an assumption that most genes don't change" (see post at https://support.bioconductor.org/p/61604/ for details). This reviewer suggest that the authors provide some data confirming that the normalization is stable enough between samples, hopefully showing that the core of genes that do not change expression reflect the size factors estimated by DEseq2 (a good example of this quality control is given in the blog post above). If some striking differences are seen, I would suggest applying HMM-normalization or other technique to better normalize the data and increase accuracy. Example of normalization methods for such situation are given in this Review by Liu et al. Front. Bioeng. Biotechnol. 2019 – DOI 10.3389/fbioe.2019.00358.

3. A known problem in high-throughput analysis is the establishment of arbitrary thresholds or cutoffs when analyzing differential expression. Here the authors often established arbitrary thresholds/cutoffs without depicting the reason for doing so (e.g., top 20 genes, 5-fold difference or 2-fold difference, etc). This reviewer thinks that it would be of great improvement for this manuscript if authors explain in detail the relevance of their choices and how it impacted their analyses.

4) Although some studies already analysed gene expression in the different organs of Aedes mosquitoes and changes occurring after a blood meal, this study is the first to analyse gene expression in most of female tissues allowing an accurate comparison of profiles between tissues. The analysis of data is very thorough and well described, showing investment (number of transcripts) and output (number of transcripts balanced by total number of transcripts in a specific tissue) of each organ. The putative biological functions of the different organs are not new and surprising, however, the gene expression profiles and conferred biological functions of the different gut regions is original and was not previously assessed. Although this study brings a lot of information and is very valuable for the mosquito research community, it remains a very descriptive study without functional characterization/validation.

Some limits should be taken in consideration when looking at the data for instance to know in which tissue a gene is expressed or not, if a gene promoter could be used for specific gene expression system, e.g., one mosquito strain analysed, immune gene expression profiles (especially in the gut) may be affected by the microbiota, which is known to be different between labs, mating status (not controlled in the study), etc. However, this study is a good start for the creation of an atlas for Aedes aegypti and should constitute the basis for a future and larger deposition of data to complete the picture, e.g., more tissues (hemocytes for instance), more developmental stages, infection with pathogens etc. The creation of an online database is of course positive, but it is regrettable that those data are not integrated in larger databases such as vector base allowing an integrated analysis of data with previous published datasets.

5) The use of cumulative expression values and RNA yield is confusing. There is not an adequate description of library prep to assess the validity of these methods. Typically, libraries are prepped with a subset of the total RNA extracted from a sample. In the manuscript, authors write that they normalized transcript levels by RNA yield, however there is no discussion of whether the volume of tissue for each replicate was standardized. If libraries were prepped with equivalent concentrations of RNA, this normalization is unnecessary. Authors may have used the concentration of RNA used to prepare the RNA-seq libraries for this normalization but the way it is written does not make their methods clear.

6) Additionally, there are points throughout the manuscript that need further explanation to improve clarity and to allow for successful assessment of the methods used. For example, when describing the mosquito equation developed by the authors, a "scaling factor" is mentioned without proper explanation of what this scaling factor is. Furthermore, specifics on the parameters used in the differential expression analysis and the gene ontology enrichment analysis are missing from the methods section.

---

## [Author Response]

Essential revisions:1. As a dry/wet bioinformatician/molecular biologist, this reviewer acknowledges that this work has been beautifully executed following standard practices for mosquito rearing and following the guidelines of RNA sequencing. Though, data reproducibility is a concern in bioinformatics since different analysis pipelines may lead to dissimilar results from same datasets. While aiming to reach a broad audience of specialists and non-specialists in RNAseq analyses, this reviewer suggests that authors deposit their command lines and scripts (even if only relevant parts) in an open-access repository such as GitHub or equivalent, so others may compare their own data without technical variation. Also, raw tables with normalized gene expression (i.e., outputs of DEseq2) could also be provided for future reference, either deposited in GEO/NCBI or as tables together with the code at GitHub.

We thank the reviewer for their kind words concerning our work. We have done as requested. Our scripts and DEseq2 outputs have been deposited at GitHub and are accessible here: https://github.com/bingdun/RNAseq_analysis_of_Aedes_aegypti.

2. One major concern relates to the normalization of transcriptomic data presented here. Authors emphasize that data from some tissues such as ovaries or specific midgut regions showed considerably distinct patterns of gene expression. Consequently, such patterns of unbalanced transcriptional levels can skew normalization and could be of concern. The tool used here for normalization of transcript counts, DEseq2, should not be severely impacted by these events since, as mentioned by its author Michael Love, "the median ratio normalization in DESeq2 doesn't have as strong of an assumption that most genes don't change" (see post at https://support.bioconductor.org/p/61604/ for details). This reviewer suggest that the authors provide some data confirming that the normalization is stable enough between samples, hopefully showing that the core of genes that do not change expression reflect the size factors estimated by DEseq2 (a good example of this quality control is given in the blog post above). If some striking differences are seen, I would suggest applying HMM-normalization or other technique to better normalize the data and increase accuracy. Example of normalization methods for such situation are given in this Review by Liu et al. Front. Bioeng. Biotechnol. 2019 – DOI 10.3389/fbioe.2019.00358.

We have extracted size factors for each of the samples in the body-part vs whole body DEseq2 analyses performed (a) with all genes in the genome and (b) with the subset of genes that were not significantly different in the first analysis (p > 0.05). We found the values were nearly identical. Since size factors are median values and, in all comparisons between body parts and whole body, fewer than half of all genes were differentially expressed, this result was consistent with expectations. In Author response image 1 we plot the two sets of size factors we extracted:

**Author response image 1. sa2fig1:** 

3. A known problem in high-throughput analysis is the establishment of arbitrary thresholds or cutoffs when analyzing differential expression. Here the authors often established arbitrary thresholds/cutoffs without depicting the reason for doing so (e.g., top 20 genes, 5-fold difference or 2-fold difference, etc). This reviewer thinks that it would be of great improvement for this manuscript if authors explain in detail the relevance of their choices and how it impacted their analyses.

We employed a variety of arbitrary cutoffs in this manuscript. These include:

a) Putative markers, Figure 1B (TPM > 5, 50x enriched versus all other body parts)

The putative markers in our manuscript serve two purposes: first, they are a suggested starting place for researchers who may want to build tissue-specific expression systems. Second, they were used as terms in the calculation of the “mosquito equation” which estimates what fraction of transcripts each body part contributes to the mosquito whole body.

We chose the >5 TPM cutoff because, for a gene to be useful as a potential tissue-specific driver, it would need to be robustly expressed. Our enrichment cutoff (50x enrichment) was chosen because we wanted to set the threshold somewhere that allowed us to identify a robust cohort of body-part specific transcripts in each body part to use in calculating the “mosquito equation”. At TPM >5 and 50x enrichment, the abdomen had the smallest number of qualifying genes (8). At the next most restrictive threshold we tried (100x) the number of qualifying genes in the abdomen fell to 3. We felt it was not desirable to base the calculation on such a small cohort.

After calculating the mosquito equation, we used it to predict the expression of each gene in the genome in whole body and plotted it against observed values (2 supp 2A). The resulting strong correlation satisfied us that 50x was a sufficiently restrictive cutoff for this purpose.

b) ‘Top 20’ (Treemaps, figures 1C, 2C, and 7E)

The purpose of the ‘Top 20’ figures was to demonstrate (a) that some transcriptomes (*e.g.*, gut) invest heavily in a few genes while others (*e.g.*, ovaries) spread their transcripts over more of the genome and (b) to illustrate the types of function that the top expressed genes in each body part have (e.g. peptidases dominating the gut, AMPs in the prov, etc.).

We chose to use the 20 highest-expressed genes because we found this was the threshold that best illustrated the contrast between transcriptomes: the top 20 genes contribute ~50% of all gut transcripts and ~10% of all ovary transcripts. We found that using more than 20 genes was not desirable as the cells in the treemaps became too small to distinguish.

c) GO plots: padj <0.05, 5x Enrichment (1C, 2B, 6 supp 1C); 2x Enrichment (1C (ovaries only) 3B, 4C)

For all GO analyses, we found, through trial and error, that a test set of ~200-1000 genes was optimal to obtain meaningful and statistically significantly enriched categories against a background of ~20,000 genes in the Aedes genome. We found that test sets of >2000 genes (>10% of the genome) were too large, resulting in higher and often non-significant p values. Thresholds of 2x and 5x enrichment versus whole-body yielded the following quantities of significantly upregulated genes (overly small and overly large test sets are marked with *):

**Author response table 1. sa2table1:** 

	2x	5x
Head	*2664	1376
Thorax	*2271	761
Abdomen	1871	359
Gut	*2163	985
MT	*2567	1193
Ovaries	555	*44
Crop	*2501	1427
Prov	1234	414
Ant	757	188
Post	*3	*1
Hind	1449	620

We found that the 5x enrichment netted a desirable fraction of the genome in all body part and gut region comparisons except for ovaries vs whole body, which yielded only 44 genes, and posterior gut versus whole gut, which yielded a single gene.

The low number of genes in ovaries meeting the 5x threshold motivated the choice to use a 2x cut-off for ovaries only. We felt this choice was justified by the fact that ovaries are the greatest contributor (~29%) of transcripts to whole gut, making it more difficult for differentially expressed genes in this organ to meet the 5x threshold.

A different approach was required for the posterior midgut which contributes ~90% of whole gut transcripts (see Figure 2E). In effect, a comparison of posterior midgut to whole gut was nearly equivalent to comparing the posterior midgut to itself. To evaluate enrichment of function in this region, we created a test set which included all genes that were significantly enriched in the posterior midgut (padj < 0.05) in comparison to all other gut regions, in a combined DESeq2 comparison, with all replicates from other regions receiving equal weight in the analysis. This yielded a test set of 608 genes. We would like to note here that in the previous version of the manuscript (and pre-print) we mistakenly stated that the GOEA for posterior midguts in *Aedes* and *Anopheles* employed a 5x cutoff. This is incorrect. No fold-change cutoff was employed for this region in either species. We regret the error and have corrected it in all figure legends and supplementary materials.

The comparisons of blood-fed timepoints and regions did not yield as many 5x enriched genes as the body part/gut region comparisons. This is unsurprising, as we were comparing differences in condition rather than in tissue. For these comparisons, we opted to use a 2x threshold.

**Author response table 2. sa2table2:** 

	**2x**	**5x**
**Upregulated**		
6 vs SF	828	228
24 vs 6	911	381
48 vs 24	469	*55
48 vs SF	102	*1
BF post vs Post	989	409
BF ant vs Ant	317	*95
**Downregulated**		
6 vs SF	1048	371
24 vs 6	741	170
48 vs 24	569	268
48 vs SF	*85	*10
BF post vs Post	737	135
BF ant vs Ant	525	108

Note: in the preprint version of this manuscript, we employed a 400-gene cutoff to limit test set size only for blood-fed comparisons. However, in the interests of consistency, we have decided to eliminate this threshold. The resulting changes in the outcome of the analysis have been disseminated to figures 3 and 4, to supplemental Table S2, and to the text of the Results section.

4) Although some studies already analysed gene expression in the different organs of Aedes mosquitoes and changes occurring after a blood meal, this study is the first to analyse gene expression in most of female tissues allowing an accurate comparison of profiles between tissues. The analysis of data is very thorough and well described, showing investment (number of transcripts) and output (number of transcripts balanced by total number of transcripts in a specific tissue) of each organ. The putative biological functions of the different organs are not new and surprising, however, the gene expression profiles and conferred biological functions of the different gut regions is original and was not previously assessed. Although this study brings a lot of information and is very valuable for the mosquito research community, it remains a very descriptive study without functional characterization/validation.Some limits should be taken in consideration when looking at the data for instance to know in which tissue a gene is expressed or not, if a gene promoter could be used for specific gene expression system, e.g., one mosquito strain analysed, immune gene expression profiles (especially in the gut) may be affected by the microbiota, which is known to be different between labs, mating status (not controlled in the study), etc. However, this study is a good start for the creation of an atlas for Aedes aegypti and should constitute the basis for a future and larger deposition of data to complete the picture, e.g., more tissues (hemocytes for instance), more developmental stages, infection with pathogens etc. The creation of an online database is of course positive, but it is regrettable that those data are not integrated in larger databases such as vector base allowing an integrated analysis of data with previous published datasets.

We agree with the reviewer that readers will need to take stock of any differences in strain, etc. when making use of our data set. The clarifications in Methods (lines 938-940) about mating status and the new supplemental table (Table S9) clarifying the details of dissections should help readers to establish the comparability of their own data/methods. We have also already noted that microbiota may affect immune gene expression in the gut (lines 805-806).

We are in contact with George Christophides and Gloria Giraldo-Calderon regarding integrating our data sets into the VectorBase transcript expression data visualization tool. Dr. Giraldo-Calderon has taken receipt of our SRA files, and forecasts that the data will be available to users as early as June of 2022.

5) The use of cumulative expression values and RNA yield is confusing. There is not an adequate description of library prep to assess the validity of these methods. Typically, libraries are prepped with a subset of the total RNA extracted from a sample. In the manuscript, authors write that they normalized transcript levels by RNA yield, however there is no discussion of whether the volume of tissue for each replicate was standardized. If libraries were prepped with equivalent concentrations of RNA, this normalization is unnecessary. Authors may have used the concentration of RNA used to prepare the RNA-seq libraries for this normalization but the way it is written does not make their methods clear.

After reviewing our Methods section, we believe we see the source of the confusion. RNA yield analyses were not performed as a correction for differences in sequencing depth between RNAseq samples (which were already normalized by the calculation of TPM). Rather, RNA yield experiments were conducted completely independently from RNAseq experiments. The RNA yield experiments serve a similar purpose (for gut regions) to the mosquito equation calculation (for body parts). The objective was to estimate what proportion of the whole gut transcriptome was derived from each gut region.

We have expanded this section of Methods to make our approach clearer (see lines 964 to 981 in the revised manuscript), but we will also briefly reiterate it here:

In RNAseq experiments: the samples extracted for the creation of libraries were diluted to the recommended concentration for library preparation, and the resulting libraries were pooled in approximately equal concentration to ensure equal depth of sequencing. Therefore, the resulting quantity of reads from each sample *does not* reflect the RNA yield of the gut regions from which they were extracted. The conversion of counts to TPMs then normalized the data across samples, correcting for any residual variation from dis-equal sequencing depth.

The RNA yield experiments were conducted by dissecting a known number of gut regions (20 to 50 for small ones, 10 for posterior guts) into 1 mL of Trizol, performing a standard phenol-chloroform extraction (no column purification), resuspending the resulting RNA pellet in a known volume, measuring the concentration by Qubit, and calculating the yield in ng/region (concentration * suspension volume / # of dissected regions).

The proportional expression of digestive enzymes/transporters (2F, 4F) and of immune genes (5E) was calculated by weighting the cumulative TPM from each category for each region (as determined by RNAseq) by that region’s RNA yield (as determined by Qubit measurement) to establish an estimate of what proportion of transcripts in the whole gut belonging to that category were derived from each contributing region.

6) Additionally, there are points throughout the manuscript that need further explanation to improve clarity and to allow for successful assessment of the methods used. For example, when describing the mosquito equation developed by the authors, a "scaling factor" is mentioned without proper explanation of what this scaling factor is.

The “scaling factor” for each body part is the proportion of transcripts in the whole body contributed by that part. For example, the head is the source of approximately 4.7% of all transcripts in the whole body, therefore its scaling factor is 0.047. The “mosquito equation” uses the scaling factors for the six body parts to predict the expression of a gene in the whole body, given TPM values for all body parts:

WB_TPM_ = Head_TPM_*0.047 + Thx_TPM_*0.194 + Abd_TPM_*0.262 + Gut_TPM_*0.14 + MT_TPM_*0.017 + Ov_TPM_*0.288

We have updated the ‘mosquito equation’ section of methods (lines 1014-1017) to more clearly explain the nature of the scaling factor (see bolded words below):

“If X is defined as the expression of a specific gene in a given body part (in TPM), and the part’s initial stands for a scaling factor equal to the proportion of transcripts in the whole body derived from that part, the relationship between the expression of that gene in each body part versus in the whole body can be expressed as...”

7) Furthermore, specifics on the parameters used in the differential expression analysis and the gene ontology enrichment analysis are missing from the methods section.

We have now deposited our scripts for topGO and DESeq2 with GitHub (see major revision #1). For all topGO analyses, we used the *p* value calculated using the classic Fisher method.